# OFFLINE MULTI-AGENT REINFORCEMENT LEARNING VIA SEQUENTIAL SCORE DECOMPOSITION

## ABSTRACT

Offline cooperative multi-agent reinforcement learning (MARL) faces unique challenges due to the distribution shift between online and offline data collection. While online MARL typically converges to a single coordinated joint policy, offline datasets are often mixtures of diverse cooperative behaviors, resulting in highly multimodal joint behavior distributions. In such settings, independent policy regularization often misaligns joint policy contraints and leads to severe distribution shift. To address this, we propose OMSD, which sequentially decomposes the joint behavior policy into individual conditional distributions and leverages diffusion-based generative models to provide modality-coordinated regularization for each agent. Combined with centralized critic guidance, OMSD achieves coordinated exploration within high-value, in-distribution regions, and avoids out-of-distribution joint actions. Experiments across multiple datasets on various continuous control tasks demonstrate that OMSD consistently achieves state-of-the-art performance, especially in challenging multimodal scenarios. Our results highlight the necessity of modality-aware coordination for robust offline MARL.

## 1 INTRODUCTION

Multi-Agent Reinforcement Learning (MARL) has achieved remarkable success in complex decision-making scenarios, including games (Berner et al., 2019; Zhang et al., 2021a), AI-driven economic models (Zheng et al., 2020), power systems (Chen et al., 2021), and traffic control (Ma et al., 2024). Yet online MARL often suffers from poor sample efficiency and a pronounced sim-to-real gap, as simulators fail to capture full complexities in the real-world and real-world exploration is risky and costly. These limitations have motivated offline MARL, which learns coordinated policies from fixed datasets without interacting with the environment during training (Yang et al., 2021; Formanek et al., 2024a). In offline MARL, a central challenge is the distribution shift problem, stemming from the disparity between the learned policy and the data collection policy (Pan et al., 2022; Barde et al., 2023). Beyond the challenges seen in single-agent offline RL (Levine et al., 2020; Prudencio et al., 2023), offline MARL must contend with exponentially large joint state-action spaces, as well as the need for high-quality coordination among agents to achieve common goals. All these challenges make effective policy learning in offline settings very difficult.

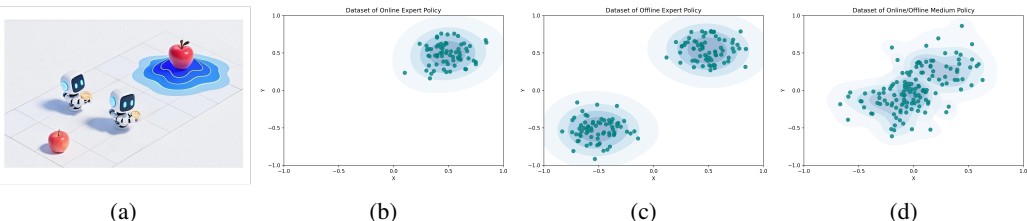

| (a) | (b) | (c) | (d) |

Figure 1: (a) Both robots need to cooperatively pick the same one of the two apples in order to receive a reward and end the game. There are two optimal strategies in this game. (b) The online expert policy converges to the optimal joint policy for either single mode due to policy dependence. (c) Offline expert datasets exhibits multi-modal optimal joint strategies due to diverse data collection sources. (d) Lower quality datasets demonstrate more pronounced multimodality.

To address these challenges, existing offline MARL methods mainly fall into two categories. The first category comprises value-based methods that build on Individual-Global-Maximization (IGM) decompositions (Rashid et al., 2018), typically coupled with conservative value estimation to mitigate critic overestimation problems under limited data coverage (Yang et al., 2021; Pan et al., 2022; Wang et al., 2023a). While these approaches alleviate extrapolation and achieve credit assignment under the Centralized-Training-Decentralized-Execution (CTDE) framework (Yang & Wang, 2020), the individual $\epsilon$-greedy policy of each agent can still lead to the selection of out-of-distribution (OOD) joint actions, which are often of low quality and may not be covered by the datasets (Matsunaga et al., 2023). The second category constrains policies via behavior-regularized updates or generates trajectories with centralized planners and world models (Matsunaga et al., 2023; Barde et al., 2023; Zhu et al., 2024). Although these methods aim to avoid OOD joint action selection through direct policy constraints, they often rely on local independent regularization for each agent. In cases where dataset policies exhibit substantial behavioral diversity, such local constraints can cause misaligned policy updates at the individual agent level, ultimately hindering the coordination required for an effective joint policy. Furthermore, centralized planners introduce additional burdens in practice, as they often entail high inference costs and require opponent modeling, which may be imprecise (Foerster et al., 2017; Yu et al., 2022), to facilitate the translation into decentralized execution strategies for each agent.

From the perspective of data distribution, the fundamental cause of these limitations lies in the stark difference between online and offline MARL data collection, as exemplified by a simple 2-agent cooperative harvesting task (Fig. 1). This is a common game with multiple Nash Equilibria, where the optimal strategy is for both players to go together to either of the apples. Online MARL resolves this ambiguity via interactive, on-policy adaptation: coupled updates and exploration break symmetry and drive convergence to a single equilibrium, yielding a single-mode joint policy. In contrast, offline MARL datasets are typically mixtures collected from diverse sources with various cooperative policies (Formanek et al., 2023; 2024a), demonstrating highly multimodal behavior. In such scenarios, the multiplicative decomposition of joint policies commonly used in online MARL can lead to biased regularization across agents, as it fails to account for the dependencies introduced by multimodality. Consequently, each agent may be pulled toward different modes, resulting in a misaligned joint policy that lies outside the high-density regions of the dataset.

In this paper, we propose the **Offline MARL with Sequential Score Decomposition** (OMSD) method to achieve coordination regularization under multimodal joint behavior policies. In particular, OMSD sequentially factorizes the joint behavior policy into individual conditional behavior distributions conditioned on both states and prefix-actions, providing an unbiased reference for each agent's Kullback–Leibler (KL) divergence policy constraints. Then the flexible diffusion models are trained to capture complex individual conditional distributions of each agent and estimate the action-space gradient of the KL constraints with score functions (Song et al., 2020a). Finally, OMSD combines the individual scores with the centralized critic gradient to guide appropriate exploration within the modality and reduce extrapolation bias with limited data coverage. This design ensures modality-consistent coordination regularization without explicit access to the full joint policy, and guides to high-value in-distribution regions without OOD joint action selection problems. Extensive experiments across various datasets and continuous control tasks demonstrate that OMSD significantly outperforms existing methods, notably excelling in multimodal scenarios such as medium datasets.

In summary, our contributions are threefold: (i) We identify the multimodal behavior policy distribution introduced by the online-offline data collection gap as the root cause of the difficulty in offline MARL policy coordination, and shed light on how independent regularization can misalign agents and cause the joint action policy distribution to shift; (ii) We develop OMSD, which sequentially decomposes behavior policies and learns diffusion-based conditional scores as a behavior regularizer, which can guarantee coordinated mode selection without modeling a full joint policy or relying on a planner; (iii) We demonstrate state-of-the-art performance on a multi-agent continuous control task benchmark, effectively handling scenarios with multimodal data distribution.

## 2 PRELIMINARIES

### 2.1 PARTIALLY OBSERVABLE STOCHASTIC GAME

A partially observable stochastic game (POSG; Hansen et al., 2004) or Markov game is defined as a tuple: $\langle \mathcal{X}, \mathcal{S}, \{\mathcal{A}^i\}_{i=1}^n, \{\mathcal{O}^i\}_{i=1}^n, \mathcal{P}, \mathcal{E}, \{\mathcal{R}^i\}_{i=1}^n \rangle$, where $n$ is the number of agents, $\mathcal{X}$ is the agent space, $\mathcal{S}$ is a finite set of states, $\mathcal{A}^i$ is the action set for agent $i$, $\boldsymbol{\mathcal{A}} = \mathcal{A}^1 \times \mathcal{A}^2 \times \cdots \times \mathcal{A}^n$ is the set of joint actions, $\mathcal{P}(s'|s, \boldsymbol{a})$ is the state transition probability function, $\mathcal{O}^i$ is the observation set for agent $i$, $\boldsymbol{\mathcal{O}} = \mathcal{O}^1 \times \mathcal{O}^2 \times \cdots \times \mathcal{O}^n$ is the set of joint observations, $\mathcal{E}(\boldsymbol{o}|s)$ is the emission function, and $\mathcal{R}^i : \mathcal{S} \times \boldsymbol{\mathcal{A}} \times \mathcal{S} \to \mathbb{R}$ is the reward function for agent $i$. The game progresses over a sequence of stages called the *horizon*, which can be finite or infinite. This paper focuses on the episodic infinite horizon problem, where each agent aims to minimize the expected discounted cumulative cost.

In a cooperative POSG (Song et al., 2020b), the relationship between agents $x$ and $x'$ is given by:

$$\forall x \in \mathcal{X}, \forall x' \in \mathcal{X} \setminus \{x\}, \forall \pi_x \in \Pi_x, \forall \pi_{x'} \in \Pi_{x'}, \frac{\partial \mathcal{R}^{x'}}{\partial \mathcal{R}^x} \geqslant 0,$$

where $\pi_x$ and $\pi_{x'}$ are policies in the policy spaces $\Pi_x$ and $\Pi_{x'}$, respectively. The inequality condition intuitively means that there is no conflict of interest among any pair of agents. The paper addresses the fully cooperative POSG, also known as the decentralized partially observable Markov decision process (Dec-POMDP; Bernstein et al., 2002), where all agents share the same global cost at each stage, i.e., $\mathcal{R}^1 = \mathcal{R}^2 = \cdots = \mathcal{R}^n$. The optimization goal for Dec-POMDP is defined as: $\min_\Psi \sum_{i=1}^n \sum_{t=0}^\infty \mathbb{E}_{s_0 \sim p_0, \boldsymbol{o} \sim \mathcal{E}, a \sim \boldsymbol{\pi}_\Psi} [\gamma^t r_{t+1}^i]$ where $\Psi := \{\psi^i\}_{i=1}^n$ are the parameters of the approximated policies $\pi_{\psi^i}^i : \mathcal{O}^i \to \mathcal{A}^i$, and $\boldsymbol{\pi}_\Psi := \prod_{i=1}^n \pi_{\psi^i}^i$ is the joint policy of all agents. Here, $\gamma$ is the discount factor, $p_0$ is the initial state distribution, and $r_{t+1}^i$ is the reward received by agent $i$ at timestep $t + 1$ after taking action $a_t^i$ in observation $o_t^i$. In the offline setting, we only have a static dataset of transitions $\mathcal{D} = (o_t^m, a_t^m, o_{t+1}^m, r_t^m)_{m=1}^{nk}$, where $k$ is the number of transitions for each agent.

### 2.2 DIFFUSION PROBABILISTIC MODELS

Diffusion probabilistic models (Sohl-Dickstein et al., 2015; Ho et al., 2020) are a likelihood-based generative framework designed to learn data distributions $q(\boldsymbol{x})$ from offline datasets $\mathcal{D} := \boldsymbol{x}^i$, where $i$ indexes individual samples (Song, 2021). A key feature of these models is the representation of the (Stein) score function (Liu et al., 2016), which does not require a tractable partition function.

The model's discrete-time generation procedure involves a forward noising process, defined as $q(\boldsymbol{x}_{k+1}|\boldsymbol{x}_k) := \mathcal{N}(\boldsymbol{x}_{k+1}; \sqrt{\tilde{\alpha}_k}\boldsymbol{x}_k, (1-\tilde{\alpha}_k)\boldsymbol{I})$, at diffusion timestep $k$. This is paired with a learnable reverse denoising process, $p_\theta(\boldsymbol{x}_{k-1}|\boldsymbol{x}_k) := \mathcal{N}(\boldsymbol{x}_{k-1}|\mu_\theta(\boldsymbol{x}_k, k), \Sigma_k)$, where $\mathcal{N}(\mu, \Sigma)$ represents a Gaussian distribution with mean $\mu$ and variance $\Sigma$. The variance schedule is defined by $\alpha_k \in \mathbb{R}$. In this framework, $\boldsymbol{x}_0 := \boldsymbol{x}$ corresponds to a sample in $\mathcal{D}$, and $\boldsymbol{x}_1, \boldsymbol{x}_2, \ldots, \boldsymbol{x}_{K-1}$ are latent variables, with $\boldsymbol{x}_K \sim \mathcal{N}(\boldsymbol{0}, \boldsymbol{I})$ for appropriately chosen $\tilde{\alpha}_k$ values and a sufficiently large $K$.

Starting with Gaussian noise, samples are iteratively generated through a series of denoising steps. The training of the denoising operator is guided by an optimizable and tractable variational lower bound, with a simplified surrogate loss proposed in Ho et al. (2020):

$$\mathcal{L}_{\text{denoise}}(\theta) := \mathbb{E}_{k \sim [1,K], \boldsymbol{x}_0 \sim q, \epsilon \sim \mathcal{N}(\boldsymbol{0}, \boldsymbol{I})} \left[ \|\epsilon - \epsilon_\theta(\boldsymbol{x}_k, k)\|^2 \right] \tag{1}$$

Here, the predicted noise $\epsilon_\theta(\boldsymbol{x}_k, k)$, parameterized by a deep neural network, approximates the noise $\epsilon \sim \mathcal{N}(\boldsymbol{0}, \boldsymbol{I})$ added to the dataset sample $\boldsymbol{x}_0$ to produce the noisy $\boldsymbol{x}_k$ in the noising process.

### 2.3 POLICY BASED OFFLINE RL

Policy based methods are successful and widely used in the offline RL algorithm community. Prior work (Nair et al., 2020) formulates the offline policy optimization problem as:

$$\max_\pi \mathbb{E}_{s \sim \mathcal{D}_\mu} \left[ \mathbb{E}_{a \sim \pi(s)} [Q_\phi(s, a)] - \frac{1}{\beta} \mathcal{D}_{\text{KL}} (\pi(\cdot|s) \| \mu(\cdot|s)) \right], \tag{2}$$

where $Q_\phi(s, a)$ is a neural network approximation of the state-action value functions $Q^\pi(s, a) := \mathbb{E}_{s_t=s, a_t=a; a_{t+1} \sim \pi} [\sum_{t=0}^{\infty} \gamma^t r(s_t, a_t)]$ under the current policy $\pi$, and $\beta$ is temperature coefficient to control how far the learned policy derive from the behavior policy $\mu$. The closed form solutions for this optimization problem (2) has been proved as

$$\pi^*(a \mid s) = \frac{1}{Z(s)} \mu(a \mid s) \exp\left(\beta Q_\phi(s, a)\right),$$

where $Z(s)$ is the partition function. A subsequent challenge is to efficiently distill the optimal policy into a parameterized policy $\pi_\theta$. A common approach is minimizing the KL-divergence between $\pi_\theta$ and $\pi^*$ with either forward or reverse direction (Chen et al., 2024). While the optimal policy may be multi-modal, meaning it has multiple equivalent policy mode distributions, it is not necessary to express every policy mode explicitly during execution. Therefore, it is a suitable choice to leverage the natural of mode-seeking characteristic in reverse-KL and capture one feasible modal in the parameterized policy with a simple distribution like Gaussian policy or deterministic policy.

**Lemma 2.1** (Behavior-Regularized Policy Optimization (BRPO) (Wu et al., 2019)). *In policy-based offline RL, given an optimal policy $\pi^*$ and a parameterized policy $\pi_\theta$, the policy regularization learning objective with reverse KL-divergence can be written as,*

$$\min_\theta \mathbb{E}_{s \sim \mathcal{D}_\mu} \underbrace{D_{KL}\left[\pi_\theta(\cdot|s) \| \pi^*(\cdot|s)\right]}_{\text{Reverse KL}} \Leftrightarrow \max_\theta \underbrace{\mathbb{E}_{s \sim \mathcal{D}_\mu, a \sim \pi_\theta} Q_\phi(s, a) - \frac{1}{\beta} D_{KL}\left(\pi_\theta(\cdot|s) \| \mu(\cdot|s)\right)}_{\text{Behavior-Regularized Policy Optimization}}. \quad (3)$$

## 3 METHODOLOGY

### 3.1 JOINT BEHAVIOR POLICY FACTORIZATION MISMATCH IN OFFLINE MARL

The multi-modality of joint behavior policy distributions in offline MARL arises from several key factors. First, many cooperative games admit multiple joint policies with similar quality, which is the notorious multiple Nash equilibrium problem. This yields datasets with diverse but equally effective behaviors, complicating policy learning. Second, in large-scale multi-agent systems, especially with homogeneous agents, data collection often anonymizes agent identities (Franzmeyer et al., 2024). Even under a single joint policy, agent trajectories become indistinguishable due to agent interchangeability, introducing inherent symmetry and multi-modality. Furthermore, offline datasets are often constructed by mixing demonstrations from various expert and suboptimal strategies due to the high cost of data collection, further increasing behavioral diversity.

Despite this evidence, a common pitfall in offline policy-based methods is the policy factorization assumption, which posits that the joint behavior policy can be factorized as $\mu(\boldsymbol{a}|s) = \prod_{i=1}^{n} \mu_i(a_i|s)$. For example, AlberDICE (Matsunaga et al., 2023, Eq. 4) implements an occupancy measure penalty using a factorized model $d^D(s, a_i)\pi_{-i}^D(\boldsymbol{a}_{-i}|s, a_i)$, where $-i$ represents all agents except agent $i$, thereby regularizing each agent based on its own marginal behavior and effectively assuming conditional independence. Similarly, DOM2 (Li et al., 2023) trains independent diffusion models for each agent based on local behavioral data, which presupposes that joint behavior can be recovered from marginal distributions. While such factorized regularization is well-motivated and effective in online settings with consistent exploration and joint update adaptation, it will lead to miscoordination of and a significant distribution shift in offline domains where the behavior policy is multimodal and strongly coupled. To formalize this issue, we analyze a stylized scenario and formulate it as a combinatorial mode mixing (CMS) proposition (proof in Appendix G.1).

**Proposition 3.1** (**Combinatorial Mode Shift (CMS)**). *Consider a fully cooperative $n$-player game with a single state and continuous action space $\mathcal{A} = [0, 1]^n$. Let $\pi^*$ be the optimal joint policy with two optimal modes: $\mathbf{a}_1 = (1, ..., 1)$ and $\mathbf{a}_2 = (0, ..., 0)$. Let $\hat\pi$ be a factorized approximation of $\pi^*$ such that $\hat\pi(\mathbf{a}) = \prod_{i=1}^{n} \hat\pi_i(a_i)$, where each $\hat\pi_i$ is learned independently. Then we have each $\hat\pi_i$ converges to Uniform$(\{0, 1\})$. The reconstruction of joint policy $\hat\pi$ exhibits $2^n$ modes, each with probability $2^{-n}$. The total variation distance between $\pi^*$ and $\hat\pi$ is:*

$$\delta_{TV}(\pi^*, \hat\pi) = 1 - 2^{1-n}$$

*As $n \to \infty$, $\delta_{TV}(\pi^*, \hat\pi) \to 1$, indicating a severe distribution shift.*

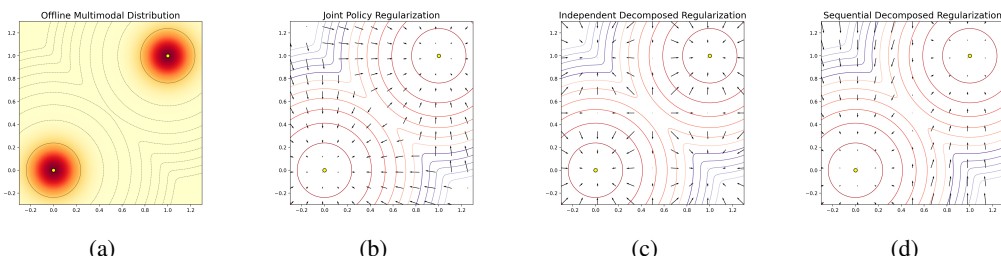

Figure 2: From left to right are (a) original multi-modal data distribution; (b) standard canonical direction of joint action; (c) biased canonical direction caused by Combinatorial Mode Shift; (d) sequential decomposition proposed by OMSD to ensure unbiased canonical direction.

This result highlights a structural failure: even though the expert policy $\pi^*$ is low-entropy and well-coordinated, the factorized approximation $\hat{\pi}$ infinitely diffuses its support set over exponentially many incoherent joint actions. Such a combinatorial mode shift arises because each agent's behavior policy $\mu_i$ is forced to match its own marginal, ignoring the inter-agent coordination property. Consequently, each agent regresses to an average over modes in its own action space, resulting in an artificial density–mode mismatch: the highest-probability joint actions under $\hat{\pi}$ may not correspond to any trajectory in the datasets.

During offline policy update, this means the behavior regularization term $\mathcal{D}_{\mathrm{KL}}(\pi_{\theta^i}(a_i|s)|\mu_i(a_i|s))$ fails as a reliable constraint: it may systematically steer policy updates toward spurious solutions disconnected from true global coordination. The recovered joint policies lose alignment with any real mode in the datasets, leading to low-efficiency exploration in areas with low data coverage regions. Specifically for the BRPO algorithm, we can summarize the biased regular coordination caused by CMS into combinatorial mode shift.

**Corollary 3.2** (Joint Policy Distribution Shift). *Let $\mu(\mathbf{a}|s)$ be a joint behavior distribution with $K$ coordinated modes over $n$ agents. When each agent regularizes to its own marginal $\mu_i(a_i|s)$ and the joint policy is factorized as $\prod_{i=1}^n \pi_i(a_i|s)$, the resulting policy exhibits probability mass on $K^n$ joint actions. As $n$ grows, the total variation distance $\delta_{TV}(\mu, \prod_i \pi_i) \to 1$, indicating a severe distribution shift from the data distribution.*

This pitfall holds whether the underlying BRPO variant is fully independent or uses a centralized critic with the CTDE framework: as long as the regularization is decomposed over agent marginals, policy updates can drift toward spurious high-density configurations unrepresentative of any valid global coordinated behavior in the data. We use a simple 2-Gaussian mixed data distribution to illustrate the regularization directions brought by different policy decomposition methods in Fig. 2.

### 3.2 SEQUENTIAL SCORE DECOMPOSITION OF JOINT BEHAVIOR POLICY

To address these limitations, we propose a novel policy learning framework named **Offline MARL with Sequential Score Decomposition** (OMSD). This method is designed to provide unbiased, coordinated, and decentralized policy updates in offline learning where joint behavior distributions $\mu(\boldsymbol{a}|s)$ are often complex and highly entangled.

Inspired by coordinate descent and rollout update (Wang et al., 2023b), we address this issue via a *sequential decomposition* of the joint behavior policy. Specifically, we model the behavior distribution as:

$$\boldsymbol{\mu}(\boldsymbol{a}|s) = \Pi_{i=1}^n \hat{\mu}_i(a_i|s, a_{<i}),$$

where $a_{<i}$ denotes the joint actions of all preceding agents, i.e., $a_{<i} = (a_1, \ldots, a_{i-1})$, with each $a_j$ sampled from the corresponding policy $\pi_j(a_j|s)$ for $j = 1, \ldots, i-1$. This sequential modeling allows each agent to learn its behavior not in isolation but conditionally on earlier agents, capturing inter-agent dependencies without requiring full joint modeling. Crucially, this structure ensures that individual policy constraints remain aligned with the joint behavior distribution, avoiding the OOD joint policies.

Figure 3: Illustration of OMSD: (Top Row) Training sequential diffusion models for each agent to distill score regularization, (Bottom Row) Plugin the sequential score models with joint action Q-gradient.

Following the BRPO framework (Chen et al., 2024), we now formulate the policy-based offline MARL under the CTDE paradigm. The goal is to learn decentralized policies $\{\pi_i(a_i|s)\}$ that maximize the joint value while remaining close to the dataset behavior:

$$\mathcal{L}^i = \min_{\theta_i} \mathbb{E}_{s \sim \mathcal{D}_\mu} D_{\mathrm{KL}}[\pi_\theta(\cdot \mid s) || \pi^*(\cdot \mid s)]$$

$$= \max_{\theta_i} \mathbb{E}_{s \sim \mathcal{D}^\mu, \boldsymbol{a} \sim \pi_\theta(\cdot|s)} Q^{tot}(s, \boldsymbol{a}) - \frac{1}{\beta} D_{\mathrm{KL}}\left[\pi_{\theta_i}(\cdot \mid \boldsymbol{s})\pi_{\theta_{-i}}(\cdot \mid \boldsymbol{s}) || \mu_i(\cdot \mid \boldsymbol{s}, a_{<i})\boldsymbol{\mu}_{-i}\right], \quad (4)$$

where $Q^{tot}(s, \boldsymbol{a})$ represents the joint state-action value estimation, $\boldsymbol{\mu}_{-i} = \prod_{j=1}^{i-1} \mu_j(a_j|s, a_{<j}) \prod_{j=i+1}^{n} \mu_j(a_j|s, a_{<j})$ denotes the conditional behavior distribution of all agents except $i$. This formulation implies the following per-agent policy gradient:

$$\nabla_{\theta_i}\mathcal{L}^i = \mathbb{E}\left[\nabla_{a_i} Q^{tot}(s, \boldsymbol{a})\big|_{a=\pi_\theta(\cdot|\boldsymbol{s})} + \frac{1}{\beta}\nabla_{a_i} \log \mu_i(\cdot \mid \boldsymbol{s}, a_{<i})\big|_{a_i=\pi_{\theta_i}(s)}\right] \nabla_{\theta_i}\pi_{\theta_i}(a_i|s), \quad (5)$$

where the expectation is taken on $s \sim \mathcal{D}^\mu, a^{-i} \sim \pi_{\theta_{-i}}$.

This gradient update allows each agent to balance between maximizing expected return and adhering to its own conditional behavior policy, conditioned on the updated actions of its prefix agents. Such bottom-up sequential guidance serves as a natural safeguard against distributional shift. Even when early agents in the sequence generate slightly OOD actions, the conditional dependency structure ensures that the current agent is updated with respect to a meaningful, in-distribution context.

### 3.3 Practical Algorithm

Clearly, the policy update gradient in equation (5) consists of a centralized Q-gradient and a gradient of an unknown logarithm probability distribution. To initialize agent policies, we first adopt centralized offline IQL to learn a joint value function, and then pretrain a conditional diffusion model for each agent, where $\hat{\epsilon}_i = \nabla_{a_i} \log \mu(a_i|s, a_{<i})$. Each agent's score model is trained using only the dataset, and the pretraining is fully parallelizable across agents, making it scalable for any team size.

Inspired by SRPO (Chen et al., 2024), instead of explicitly modeling the behavior policy distribution $\mu_i(a_i|s, a_{<i})$, we can distill agent-wise score functions $\hat{\epsilon}_i = \nabla_{a_i} \log \mu(a_i|s, a_{<i})$ from pretrained diffusion models as gradient regularization into policy update at low noise levels ($t \to 0$), efficiently providing score approximations without requiring sampling actions from consuming denoising process. This transforms policy decomposition into direction-aware regularization, effectively controlling update deviation and encouraging high-value yet conservative exploration. Formally, each agent $i$ minimizes the regularized objective from equation 4, where the practical policy gradient becomes:

$$\nabla_{\theta_i}\mathcal{L}^i_{OMSD}(\theta_i) = \mathbb{E}[\nabla_{a_i} Q_\phi(s, \boldsymbol{a}) + \frac{1}{\beta} \underbrace{\nabla_{a_i} \log \mu(a_i \mid s, a_{<i})\big|_{a_i=\pi_{\theta_i}, a_{<i}=\hat{\boldsymbol{\pi}}_{\theta_{<i}}(s)}}_{=-\hat{\epsilon}^*_i(a_i|s,t)/\sigma_t|_{t \to 0}}]\nabla_{\theta_i}\pi_{\theta_i}(s). \quad (6)$$

To compute the regularization score $\nabla_{a_i} \log \mu(a_i|s, a_{<i})$ for $\pi_i^t$, OMSD adopts a sequential update scheme during policy update, where agent $i$ conditions on prefix actions $a_{<i}^{(new)}$ sampled from the most recently updated policies $\pi_j^{(new)} j < i$ within the same iteration. Here, $a< i^{(new)}$ indicates that, for each agent $i$, the prefix actions are generated by the current versions of agents 1 to $i-1$ after their latest updates in this round. This sequential conditioning is only applied during the policy optimization process to enable coordinated learning, while all agents can still act concurrently and independently during execution. This mechanism guarantees that the score regularization directions mutually point toward in-distribution modes of the dataset. To reduce variance in these prefixes and stabilize score estimation, we use deterministic DiLac policies, which preserve expressiveness while avoiding noise amplification in continuous control tasks. Note that the sequential structure is only required during policy update, which provides flexibility for concurrent decentralized execution and parallel diffusion models pretraining. The pseudo code is available in Appendix B. For more details, refer to Appendix H.

## 4 EXPERIMENTS AND RESULTS

In this section, we evaluate the proposed method OMSD on a bandit example and the challenging high-dimensional continuous control multi-agent testbeds (MPE) (Lowe et al., 2017) and MaMuJoCo (Peng et al., 2021). We aim to address the following questions: (i) Can OMSD learn high-quality coordinated policies from sub-optimal datasets with multi-modality distribution? (ii) How do policy factorization methods, e.g., Independent Factorization and Sequential Score Decomposition, influence the policy update? (iii) Can OMSD effectively avoid OOD distribution shift problems?

**Environments.** In the bandit example, we design a 2-agent fully cooperative task where the reward function is $r_i = a_1 * a_2$ for $i = 1, 2$. The optimal rewards are achieved with joint actions $[-1, -1]$ and $[1, 1]$. MPE include 3 tasks requiring agents cooperation to conver landmarks or catch the pretrained prey opponent in a 2D environment. In MaMuJoCo, each part of a robot is modeled as an independent agent and learn optimal motions through cooperating with each other. Further details are provided in Appendix D.

**Datasets.** For bandit problem, we generate an action dataset by randomly sampling 1,000,000 times from a 2-Gaussian mixed model with mean values $\mu_0 = [0.8, 0.8], \mu_1 = [-0.8, -0.8]$ and variance $\sigma_0 = \sigma_1 = 0.3$. Considering the inconsistencies in datasets and baselines in previous research, as noted by Formanek et al. (2024b), we select three of the most well-evaluated benchmarks, the MPE datasets provided by OMAR Pan et al. (2022), and two MaMuJoCo datasets provided by OG-MARL Formanek et al. (2023) and OMIGA Wang et al. (2023c). Each dataset contains datasets of various qualities, ranging from expert to random. All offline datasets are open-sourced[1 23].

**Baselines.** In the bandit setting, to clearly compare the learning dynamics of different policy decomposition under multi-modal datasets, we extend the standard BRPO algorithm to a multi-agent version, including BRPO-JAL (joint action learning), BRPO-IND (independent learning), and BRPO-CTDE. Detailed algorithmic descriptions are provided in the Appendix G. For high-dimensional tasks, we benchmark against state-of-the-art offline MARL methods, including independent learning approaches (BC, MATD3+BC, MA-ICQ, OMAR (Pan et al., 2022)), CTDE value decomposition methods (MA-CQL (Jiang & Lu, 2021) and CFCQL (Shao et al., 2023)), and diffusion-based techniques (MADiff (Zhu et al., 2024) and DoF (Li et al., 2025)).

### 4.1 BANDIT EXAMPLES

As shown in Table 1, OMSD demonstrates performance comparable to joint action learning algorithm BRPO-JAL, outperforming independent learning and naive CTDE methods with the factorization assumption. Clearly, both BRPO-IND and BRPO-CTDE struggle with OOD joint actions like $[1, -1]$ and $[-1, 1]$. This issue is more pronounced in continuous tasks compared to discrete XOR Matrix Games in Matsunaga et al. (2023), where behavior policies with limited expressivity often struggle to capture complex multi-modal distributions (Wang et al., 2023c).

---

[1]OMAR datasets: `https://github.com/ling-pan/OMAR`

[2]OG-MARL datasets: `https://github.com/instadeepai/og-marl`

[3]OMIGA datasets: `https://cloud.tsinghua.edu.cn/d/dcf588d659214a28a777/`

Table 2: The average normalized score on offline MARL tasks with OMAR datasets. Shaded columns represent our methods. The mean and standard error are computed over 5 different seeds.

| Testbed | Task | Dataset | BC | MA-ICQ | MA-CQL | MA-TD3+BC | OMAR | CFCQL | MADiff-D | DoF-P | OMSD |
|---|---|---|---|---|---|---|---|---|---|---|---|
| MPE | Cooperative Navigation | Expert | 35.0 ± 2.6 | 104.0 ± 3.4 | 98.2 ± 5.2 | 108.3 ± 3.3 | 114.9 ± 2.6 | 112 ± 4 | 95.0 ± 5.3 | **126.3 ± 3.1** | 102.3 ± 1.4 (-22.1%) |
| | | Medium | 31.6 ± 4.8 | 29.3 ± 5.5 | 34.1 ± 7.2 | 29.3 ± 4.8 | 47.9 ± 18.9 | 65.0 ± 10.2 | 64.9 ± 7.7 | 60.5 ± 8.5 | **70.1 ± 1.4** (+7.8%) |
| | | Random | -0.5 ± 3.2 | 6.3 ± 3.5 | 24.0 ± 9.8 | 9.8 ± 4.9 | 34.3 ± 5.3 | 62.2 ± 8.1 | 6.9 ± 3.1 | 34.5 ± 5.4 | **69.8 ± 4.6** (+12.1%) |
| | Predator Prey | Expert | 40.0 ± 9.6 | 113.0 ± 14.4 | 93.9 ± 14.0 | 115.2 ± 12.5 | 116.2 ± 19.8 | 118.2 ± 13.1 | 120.9 ± 14.6 | 120.1 ± 6.3 | **161.4 ± 4.2** (+33.5%) |
| | | Medium | 22.5 ± 1.8 | 63.3 ± 20.0 | 61.7 ± 23.1 | 65.1 ± 29.5 | 66.7 ± 23.2 | 68.5 ± 21.8 | 77.2 ± 10.4 | 83.9 ± 9.6 | **137.1 ± 6.3** (+63.0%) |
| | | Random | 1.2 ± 0.8 | 2.2 ± 2.6 | 5.0 ± 8.2 | 5.7 ± 3.5 | 11.1 ± 2.8 | 78.5 ± 15.6 | 3.2 ± 4.0 | 14.8 ± 3.2 | **133.9 ± 7.4** (+70.6%) |
| | World | Expert | 33.0 ± 9.9 | 109.5 ± 22.8 | 71.9 ± 28.1 | 110.3 ± 21.3 | 110.4 ± 25.7 | 119.7 ± 26.4 | 122.6 ± 14.4 | 138.4 ± 20.1 | **163.9 ± 10.8** (+18.4%) |
| | | Medium | 25.3 ± 2.0 | 71.9 ± 20.0 | 58.6 ± 11.2 | 73.4 ± 9.3 | 74.6 ± 11.5 | 93.8 ± 31.8 | 123.5 ± 4.5 | 86.4 ± 10.6 | **160.3 ± 4.1** (+29.8%) |
| | | Random | -2.4 ± 0.5 | 1.0 ± 3.2 | 0.6 ± 2.0 | 2.8 ± 5.5 | 5.9 ± 5.2 | 68 ± 20.8 | 2.0 ± 3.0 | 15.1 ± 3.0 | **141.1 ± 5.8** (+107.5%) |
| | Average Score | | 20.6 ± 3.9 | 55.6 ± 10.6 | 49.8 ± 12.1 | 57.8 ± 10.5 | 64.7 ± 12.8 | 87.3 ± 16.9 | 68.5 ± 7.4 | 75.6 ± 7.8 | **126.7 ± 5.1** (+33.2%) |
| MaMuJoCo (210) | 2-HalfCheetah | Good | 6846 ± 574 | - | - | 7025 ± 439 | 1434 ± 1903 | - | 8246 ± 342 | - | **8619 ± 187** (+4.5%) |
| | | Medium | 1627 ± 187 | - | - | 2561 ± 82 | 1892 ± 220 | - | 2207 ± 23 | - | **2660 ± 56** (+3.9%) |
| | | Poor | 465 ± 59 | - | - | 736 ± 72 | 384 ± 420 | - | 759 ± 18 | - | **866 ± 35** (+14.1%) |
| | 2-Ant | Good | 2697 ± 267 | - | - | 2922 ± 194 | 464 ± 469 | - | 2946 ± 77 | - | 2714 ± 248 (-7.9%) |
| | | Medium | 1145 ± 126 | - | - | 744 ± 283 | 799 ± 186 | - | 1211 ± 69 | - | **1372 ± 48** (+13.1%) |
| | | Poor | 954 ± 80 | - | - | 1256 ± 122 | 857 ± 73 | - | 946 ± 66 | - | 1213 ± 95 (-3.5%) |
| | 4-Ant | Good | 2802 ± 133 | - | - | 2628 ± 971 | 344 ± 631 | - | 3080 ± 38 | - | 2844 ± 68 (-7.7%) |
| | | Medium | 1617 ± 153 | - | - | 1843 ± 494 | 929 ± 349 | - | 1649 ± 100 | - | **1942 ± 131** (+5.3%) |
| | | Poor | 1033 ± 122 | - | - | 1075 ± 96 | 518 ± 112 | - | 1295 ± 57 | - | **1477 ± 86** (+14.1%) |

Furthermore, in Fig. 2, we visualize the policy regularization gradient directions during training by sampling joint actions. Independent factorization methods such as BRPO-IND and BRPO-CTDE exhibit miscoordination among independent regularization, potentially leading to OOD joint actions. Benefiting from unbiased score

Table 1: Evaluation rewards after convergence for the toy example.

| BRPO-IND | BRPO-JAL | BRPO-CTDE | OMSD (Ours) |
|---|---|---|---|
| 0±1 | 1±0 | 0±1 | 1±0 |

decomposition and centralized critics, OMSD with sequential score decomposition can correctly identify both the reward and behavior regularization directions, thereby ensuring convergence to the optimal mode within the dataset distribution. Our results highlight OMSD's effectiveness in enforcing the policy update within the joint behavior policy distribution and improving coordination. More detailed discussion about BRPO-IND and BRPO-CTDE can be found in Appendix G.

## 4.2 HIGH-DIMENSIONAL CONTINUOUS CONTROL TASKS

We further evaluated our algorithms on more complex continuous control tasks in the `MPE` and `MaMuJoCo` suites. Table 2 shows the normalized scores of `MPE` and original scores of `MaMuJoCo` for OMSD across various datasets. The performance of the experiment results is measured by the normalized score $100 \times (S - S_{Random})/(S_{Expert} - S_{Random})$ (Pan et al., 2022). The expert and random scores for Cooperative Navigation, Predator Prey, and World are $\{516.8, 159.8\}$, $\{185.6, -4.1\}$, and $\{79.5, -6.8\}$.

OMSD surpasses the existing state-of-the-art methods on most tasks. Specifically, on datasets with the most pronounced multimodal distributions, such as medium and random datasets, our method achieves significant improvements over previous approaches, with performance closely approaching the maximization episode rewards within datasets (as shown in Appendix E). This indicates that OMSD is capable of identifying multimodal data distributions and selecting higher-quality modes. As for the two tasks where performance is relatively poor, we find that they are mainly limited by the suboptimal performance of the pre-trained centralized critic. As a result, even though the diffusion model is able to capture the multi-modal structure in the dataset, it lacks an effective reward improvement signal to guide policy update. More detailed description of hyperparameters and pretraining can be found in Appendix D.

To further compare OMSD with other diffusion-based approaches for handling multi-modality, we include two representative baselines: MADiff-D (Zhu et al., 2024), a decentralized execution variant that leverages diffusion models for trajectory planning, and DoF-P (Li et al., 2025), which employs a diffusion model as actor to generate actions by factorizing noise. Experimental results show that OMSD consistently outperforms these methods across most tasks, particularly in cooperative-competitive scenarios that require strong coordination. We attribute this advantage to the use of diffusion models as sequential decomposed score functions estimators, which more accurately capture inter-policy dependencies, enabling a more direct and fine-grained influence on policy gradient directions. We achieved significant performance improvements of 73.9% on the OMIGA (Wang et al., 2023a) dataset in Table. 3. We speculate that this is because the OMIGA environment uses

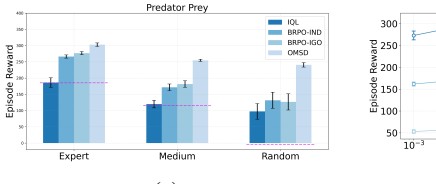 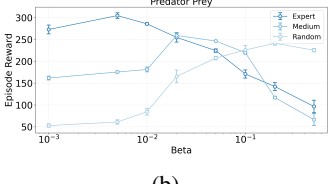 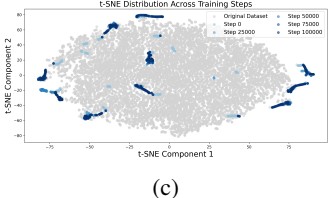

(a)   (b)   (c)

Figure 4: (a) Comparison of pre-trained IQL and post-trained algorithms. (b) Regularization term $\beta$ for OMSD performance. (c) t-SNE Visualization of policy evolution during OMSD training.

global states as observations, which facilitates more efficient behavioral regularization coordination. Additional experiment results refer to Appendix C.

### 4.3 ABLATION STUDY

**Does Score Decomposition Method Matter?** To investigate the impact of our proposed sequential score decomposition mechanism, we conduct a series of ablation studies. To keep fair comparison, we compare OMSD against BRPO-IND and BRPO-CTDE as described in Sec. F. As shown in Fig. 4a, OMSD consistently outperforms both the pretrained IQL and factorization methods, as well as the overall dataset quality. The average episode reward across datasets is indicated by a purple dashed line. The notable improvement over the pretrained IQL highlights OMSD's ability to effectively combine global critic signals with policy constraints, enabling more reliable offline policy improvement. In contrast, the performance gap between OMSD and BRPO-CTDE illustrates that inappropriate score decomposition can lead to poorly coordinated joint policies that suffer from OOD actions, ultimately degrading overall performance. The dotted lines in the figure indicate the average and maximized absolute return of the training datasets. Additionally, we verify that OMSD is insensitive to the specific agent update order, demonstrating robustness across different factorization sequences. More experiment results are provided in the Appendix D.7.1 and D.7.4.

**Hyperparameters.** Since policy-based offline methods are sensitive to the degree of behavior regularization, we conduct a systematic study on the influence of the regularization coefficient $\beta$ as shown in Fig 4b. Specifically, we sweep $\beta$ over the set $\{0.001, 0.005, 0.01, 0.02, 0.05, 0.1, 0.2, 0.5\}$. Our results show that the optimal value of $\beta$ depends strongly on the quality of the dataset: expert-level datasets benefit from stronger policy constraints (e.g., $\beta = 0.001$), preserving high-quality behaviors; in contrast, lower-quality datasets such as random favor weaker regularization (e.g., $\beta = 0.3$), allowing the policy to deviate from suboptimal demonstrations and encourage more exploratory behavior. For detailed experimental results on additional tasks, please refer to the Appendix D.7.2.

**How does OMSD avoid OOD joint actions?** We observe that OMSD achieves remarkable performance gains on low-quality datasets, where prior methods often struggle. To investigate this, we visualize the learning policy checkpoints via t-SNE (Van der Maaten & Hinton, 2008) by sampling state-action pairs from the policy and comparing them to the dataset distribution. As shown in Fig. 4c, OMSD captures the underlying multimodal structure and concentrates around high-reward regions within the dataset support. This suggests that OMSD effectively exploits the critic as a reward landmark while remaining within the data distribution, which enables stable policy improvement.

## 5 CONCLUSION

In this paper, we study the key challenge of multi-modal joint behavior policies in offline MARL and propose the sequential score decomposition algorithm OMSD with diffusion models. To our knowledge, OMSD is the first policy decomposition-based offline MARL algorithm explicitly deal the multimodal behavior policies, leveraging the decomposed score functions distilled from diffusion models to regularize the policy update gradients. Experiment results demonstrate the superiority of our methods OMSD and the effectiveness of policy improvement with coordinate action selection. One future work aims to develop more precise and optimal policy decomposition methods to enhance the ability of policy based offline MARL methods.

## 6 ETHICS STATEMENT

Our work does not involve human subjects, sensitive data, or personally identifiable information. The research is purely theoretical/empirical (choose one) and is not expected to raise any ethical concerns. All experiments are conducted in simulated environments and comply with the relevant ethical guidelines of our institution.

## 7 REPRODICIBILITY STATEMENT

We provide all the details necessary to reproduce our results. The main paper and supplementary materials contain a comprehensive description of the model architecture, training procedure, and hyperparameters. The code used to generate the main results will be publicly available on GitHub.

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

# Part I

# Supplementary Material

## Table of Contents

## A  RELATED WORKS

**Offline MARL.** Early research in offline MARL mainly made efforts to extend the pessimistic principles from offline single-agent RL with independent learning paradigm. For example, MAICQ (Yang et al., 2021) and MABCQ (Jiang & Lu, 2021) extended the pessimistic value estimation such as CQL to multi-agent and discuss the extrapolation error under exponential increasing dimension of joint actions space problem. Furthermore, OMAR (Pan et al., 2022) dealt with the local optima with zero-th order optimization. Motivated by this, CFCQL (Shao et al., 2023) further improved OMAR with counterfactual value estimation to avoid over-pessimistic value estimation. Recently, MACCA (Wang et al., 2023d) and OMIGA (Wang et al., 2023a) has incorporated causal credit assignment technique and the IGM principle into the offline value decomposition process to enhance the credit

assignment. In SIT Tian et al. (2023), authors recognized the data-imbalance problem and handle it with reliable credit assignment technique. On the other hand, AlberDICE (Matsunaga et al., 2023) and MOMA-PPO (Barde et al., 2023) recognized and addressed OOD joint action coordination problems with alternative best response and world model based planning. Our method aligns in this direction and try to model complex behavior policies with diffusion models. BRUD Tilbury et al. (2024) discusses the failure of policy updates caused by different data points under offline MADDPG-style algorithm. The prioritised dataset sampling mechanism is proposed to ensure that the sampled data in the current batch is close to the distribution of the updated policy. Although this paper considers the impact of data points on policy learning under offline MARL, MADDPG-type modeling still ignores the multimodal characteristics of the joint behavior policy distribution. Besides, there are also some works following the trajectory generation route, such as MAT (Wen et al., 2022), MADT (Meng et al., 2021), and MADTKD (Tseng et al., 2022). These methods are beyond our scope.

**Diffusion Models in RL.** Recently, motived by the great advantage of diffusion models, RL researchers turn to seek the possibilities of introducing diffusion models into RL area. Previous works can be typically divided into three topics: serving as planner, serving as policy, and serving for data augmentation. Our method mainly fall in the second topic. Single RL suffers multimodal and MLE fails due to mode cover. Diff-QL (Wang et al., 2023c) and SfBC (Chen et al., 2022) used diffusion model to represent the behavior policy and generate a batch of candidate actions with diffusion models, then use resampling to choose the executive actions. These methods suffer the inherent drawback of slow inference process of diffusion models. For this reason, some works tried to accelerate the sampling process of diffusion actor. EDP (Kang et al., 2024) and consistency-AC (Ding & Jin, 2023) leveraged the advanced diffusion models to accelerate the action sampling in RL tasks. Diff-DICE Mao et al. (2024) investigated guiding and selecting paradigm in diffusion-based RL and avoid OOD actions by proposing a guide-then-select mechanism. Recently, there are few works such as MADiff (Zhu et al., 2024) and DoF (Li et al., 2025), which take diffusion models as a centralized planner or actors. DoF (Li et al., 2025) introduces a novel diffusion-based factorization framework that explicitly models multi-agent interactions, representing significant progress in this domain. Similarly, DOM2 (Li et al., 2023) adopts diffusion models as a data augmentation tool to synthesize interaction-aware trajectories, improving cooperative behavior on shifted environments. While these works span diverse methodologies, our approach aligns with efforts to address OOD joint action challenges and complex behavior policies by leveraging advanced diffusion-based mechanisms.

# B  OMSD PSEUDO CODE

Here we provide the pseudo-code of our algorithm OMSD.

---
**Algorithm 1** OMSD Algorithm

---
1: **Input**: Offline dataset $\mathcal{D}^\mu$
2: // **Critic Pretraining**
3: **for** critic training step **do**
4:     Train centralized joint critic $Q^{tot}$
5: **end for**
6: // **Score Pretraining (Parallelizable)**
7: **for all** agent $i = 1, \ldots, n$ **in parallel do**
8:     Pretrain conditional diffusion score model $\hat{\epsilon}_i$ on $\mathcal{D}^\mu$
9: **end for**
10: // **Policy Optimization (Sequential Update)**
11: **for** policy gradient step **do**
12:     **for** agent $i = 1, \ldots, n$ (in order) **do**
13:         Sample prefix actions $a_{<i}$ using latest policies $\{\pi_j\}_{j<i}$
14:         Update $\theta_i \leftarrow \theta_i + \alpha \nabla_{\theta_i} \mathcal{L}^i_{OMSD}(\theta_i)$ (Eq. 6)
15:     **end for**
16: **end for**

---

## C ADDITIONAL EXPERIMENTS ON MAMUJOCO

In this sections, we provide additional experimental results to demonstrate the scalability and versatility of our method across different task scenarios. In Table 3, the experiment results are trained on the `MaMuJoCo` datasets provided by OMIGA (Wang et al., 2023a). OMSD significantly outperforms baselines across all tasks, achieving an impressive average improvement of 73.9%. This advantage is particularly pronounced on mixed datasets such as Medium-Expert and Medium-Replay, validating its effectiveness in modeling complex joint behavior policies. As the sequential decomposition process is only conditioned on prefix local actions rather than states, the training complexity of the diffusion model is similar for all agents. Therefore, OMSD can naturally be extended to more complex tasks with a larger number of agents, such as 6-agent HalfCheetah.

Table 3: Experiment results on the `MaMuJoCo` environments with OMIGA (Wang et al., 2023a) datasets.

| Task | Dataset | BCQMA | CQLMA | ICQ | OMAR | OMIGA | OMSD (ours) |
|------|---------|-------|-------|-----|------|-------|-------------|
| 6-HalfCheetah | Expert | 2992.71±629.65 | 1189.54±1034.49 | 2955.94±459.19 | -206.73±161.12 | 3383.61±552.67 | **5545±156 (+64%)** |
| | Medium-Expert | 3543.70±780.89 | 1194.23±1081.06 | 2833.99±420.32 | -253.84±63.94 | 2948.46±518.89 | **5237±46 (+48%)** |
| | Medium-Replay | -333.64±152.06 | 1998.67±693.92 | 1922.42±612.87 | -235.42±154.89 | 2504.70±83.47 | **4582±52 (+83%)** |
| | Medium | 2590.47±1110.35 | 1011.35±1016.94 | 2549.27±96.34 | -265.68±146.98 | 3608.13±237.37 | **4695±62 (+30%)** |
| 3-Hopper | Expert | 77.85±58.04 | 159.14± 313.83 | 754.74± 806.28 | 2.36± 1.46 | 859.63±709.47 | **3595 ± 66 (+329%)** |
| | Medium-Expert | 54.31±23.66 | 64.82±123.31 | 355.44±373.86 | 1.44±0.86 | 709.00±595.66 | **3568 ± 45 (+403%)** |
| | Medium | 44.58±20.62 | 401.27±199.88 | 501.79±14.03 | 21.34±24.90 | 1189.26± 544.30 | **3360 ± 276 (+183%)** |
| 2-Ant | Expert | 1317.73±286.28 | 1042.39±2021.65 | 2050.00±11.86 | 312.54±297.48 | 2055.46±1.58 | **2191 ± 46 (+6.6%)** |
| | Medium-Expert | 1020.89±242.74 | 800.22±1621.52 | 1590.18±85.61 | -2992.80± 6.95 | 1720.33±110.63 | **2002 ± 124 (+16.4%)** |
| | Medium-Replay | 950.77±48.76 | 234.62±1618.28 | 1016.68±53.51 | -2014.20±844.68 | **1105.13±88.87** | 1009 ± 43(-8.7%) |
| | Medium | 1059.60±91.22 | 533.90±1766.42 | 1412.41±10.93 | -1710.04±1588.98 | 1418.44±5.36 | **1619 ± 77 (+14.2%)** |
| | Average | 1210.82±313.12 | 784.56±1044.66 | 1631.17±267.71 | -667.37±299.29 | 1954.74±313.48 | **3400±90 (+73.9%)** |

Experimental results on Table 4 are trained on the 2-agent Halfcheetah dataset provided by OMAR Pan et al. (2022). In this experiment, OMSD achieves the best performance in three scenarios across four experiment settings. The most significant improvement is observed on the Medium-Replay dataset, highlighting the challenge posed by the severe multimodal distribution of joint behavior policies on mixed-quality datasets to offline MARL algorithms, which can be effectively captured and handled by our methods. Poor performance on the random-quality dataset is attributed to the difficulty of learning the centralized critic on this dataset. Furthermore, since the behavioral policies on the poor dataset are the worst, the policy regularization learned by the diffusion model struggles to provide stable policy constraints and performance improvements. This suggests that our approach may benefit from combining it with better critics from more robust value-based offline MARL training methods.

Table 4: Experiment results on the `MaMuJoCo` environments with OMAR (Pan et al., 2022) datasets.

| Task | Dataset | MA-ICQ | MA-CQL | MA-TD3+BC | OMAR | CFCQL | OMSD |
|------|---------|--------|--------|-----------|------|-------|------|
| 2-HalfCheetah | Expert | 110.6 ± 3.3 | 50.1±20.1 | 114.4 ± 3.8 | 113.5±4.3 | 118.5 ± 4.9 | **119.0 ± 1.3** (+0.4%) |
| | Medium | 73.6 ± 5.0 | 51.5±26.7 | 75.5±3.7 | 80.4±10.2 | 80.5±9.6 | **81.4 ± 7.2** (+1.2%) |
| | Med-Replay | 35.6±2.7 | 37.0±7.1 | 27.1±5.5 | 57.7±5.1 | 59.5 ± 8.2 | **78.9 ±4.4** (+32.6%) |
| | Random | 7.4±0.0 | 5.3±0.5 | 7.4±0.0 | 13.5±7.0 | **39.7±4.0** | 15.6±4.2 (-60.7%) |

## D EXPERIMENTAL DETAILS

In this section, we highlight the most important implementation details for the OMSD and baselines. More details can be found in our open-source code.

### D.1 ENVIRONMENT DETAILS

We use the open-source implementations of multi-agent particle environments[4] Lowe et al. (2017) and `MaMuJoCo`[5] Peng et al. (2021). Fig. 5 and Fig. 6 illustrate the rendered environments.

In Cooperative Navigation task, 3 learning agents need to cooperatively spread to 3 landmarks, where the common rewards are based on the distances away from landmarks with collusion penalties. In

---

[4]https://github.com/openai/multiagent-particle-envs
[5]https://github.com/schroederdewitt/multiagent_mujoco

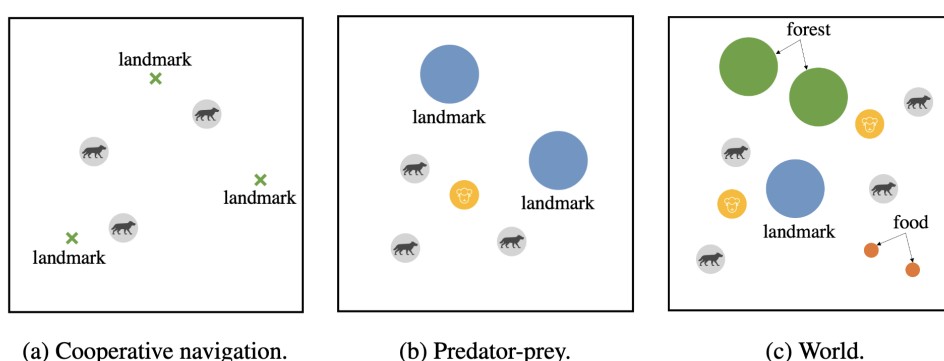

(a) Cooperative navigation.  (b) Predator-prey.  (c) World.

Figure 5: `MPE` environments. Pan et al. (2022)

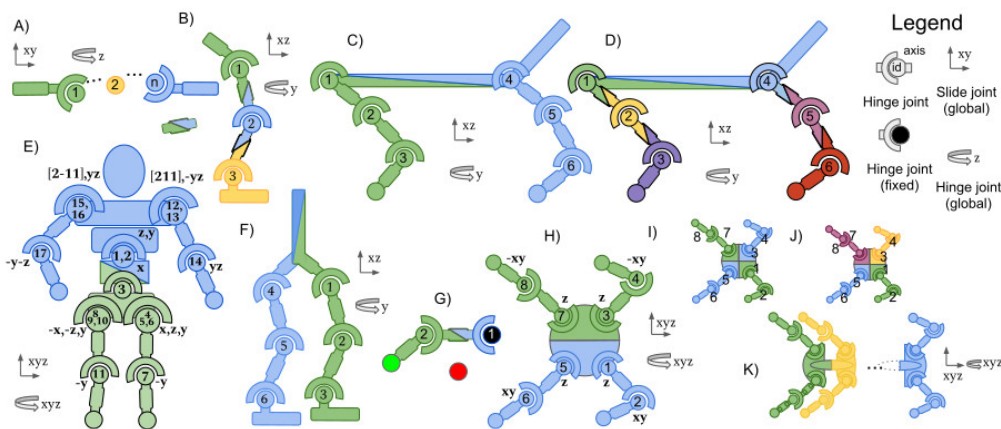

Figure 6: `MaMuJoCo` environments. (Peng et al., 2021)

Predator Prey, 3 predators are trained to catch a moving prey, which challenge the predators to surround the prey with high degree of coordination. In world, the original settings involves 4 slower cooperating predators to catch 2 faster preys, where the preys are rewarded by avoiding being captured and eating foods. However, the offline datasets provided by OMAR is trained with 3 slower predators and 1 prey. In 2-agent HalfCheetah task, a halfcheetah with 6 joints need to keep moving forward. The 6 joints are divided into two groups, where each agent controls 3 joints, representing the front legs and the hind legs respectively.

Specifically, we noticed that several commonly used datasets have different settings for `MaMuJoCo`, which affects the dimension of the observation space. Taking the `2-agent Halfcheetah` dataset as an example, the OMAR dataset uses `obsk=0`, disregarding neighbor information, resulting in a state space dimension of `state_dim=17` and an observation space dimension of `obs_dim=6`. OMIGA customizes an environment wrapper, and the returned observation variables are actually global state variables, with both the state space and observation space dimensions of `state_dim=17`. The OGMARL dataset additionally sets `obsk=1` to consider neighbor information and `global categories: {qvel, qpos}`, causing the observation space to be expanded to `obs_dim=13` dimensions. MADiff, due to its transformer structure, adds one-hot encoding to OGMARL to represent the agent ID, resulting in an observation dimension of `obs_dim=15` dimensions. To ensure fairness, this paper uniformly follows the original dataset collection process settings, removing the one-hot ID from the MADiff dataset to ensure it is independent of agent ID information. Besides, both OMAR and OMIGA employs `mujoco 200` and `mamujoco=0.0.1`, while OGMARL employs `mujoco 210` and `mamujoco=1.1.0`. The different versions of the mujoco suites will also lead to obvious performance differences.

## D.2 Baseline Settings

In this section, we provide additional details for each of the baseline algorithms. All scores of baselines are derived from the standardized scores reported in the MADiff (Zhu et al., 2024) and the DoF (Li et al., 2025). Consider that OMSD is developed as a CTDE algorithm for continuous control tasks, we select the decentralized version MADiff-D and DoF-P. The open-sourced implementations of baselines are from (Iqbal & Sha, 2019)[6], OMAR (Pan et al., 2022), CFCQL (Shao et al., 2023)[7], MADiff (Zhu et al., 2024)[8], and DoF (Li et al., 2025)[9].

## D.3 Network Architecture

The hyperparameter and network architecture settings for pre-training primarily follow those of the standard IQL algorithm Kostrikov et al. (2021) and SRPO algorithm Chen et al. (2024).

For the centralized critic model, we adapt it from the standard IQL implementation[10]. This model consists of a deterministic policy network, a state-action value network (Q-net) with double-Q learning for stabilized training, and a state value network (V-net). All networks are structured as 2-layer MLPs with 256 hidden units and ReLU activations. The deterministic policy network is optimized using annealing AdamW with a learning rate of $3 \times 10^{-4}$, while the value networks are trained using Adam with a fixed learning rate of $3 \times 10^{-4}$.

The diffusion behavior model is implemented as a 2-layer U-Net with 512 hidden units. The time embedding dimension is set to 64, and the embedding dimension for concatenated input (state and actions) is 32. The learning rate is $3 \times 10^{-4}$.

The policy model is a Dilac policy represented by a 2-layer MLP with 256 hidden units and ReLU activations. It is trained using the Adam optimizer with a learning rate of $3 \times 10^{-4}$ and a batch size of 512. The training process consists of 1.0 million gradient steps for `MaMuJoCo` tasks and 0.1 million gradient steps for `MPE` tasks.

The key hyperparameters for OMSD are summarized in Table 5.

Table 5: Hyper-Parameters for OMSD

| Algorithm | Hyper-Parameter Name | Value |
| --- | --- | --- |
| All | Batch Size | 512 |
| All | Optimizor | Adam |
| All | Learning Rate | $3 \times 10^{-4}$ |
| All | Hidden Activation Function | ReLU |
| All | Discount Factors of RL $\gamma$ | 0.99 |
| All | Soft Update Rate of Target Networks $\tau$ | 0.005 |
| All | MPE Episode Length | 25 |
| All | MaMuJoCo Episode Length | 1000 |
| All | Buffer Size | 1e6 |
| All | Reward Scale | 1 |
| Critic & Diffusion Models | Training Epochs | 200 |
| Critic & Diffusion Models | Training Steps in Each Epoch | 10000 |
| Critic & Diffusion Models | Actor Blocks | 2 |
| Critic Models | Q-Network Layers | 2 |
| Diffusion Models | Time Gaussian Projection Dims | 32 |
| Diffusion Models | Time Embedding Dims | 64 |
| Diffusion Models | State-action Embedding Dims | 32 |
| Diffusion Models | Resnet Hidden Dims | 512 |
| Diffusion Models | Dilac Policy Learning Rate | 3e-4 |

---

[6]https://github.com/shariqiqbal2810/maddpg-pytorch
[7]https://github.com/thu-rllab/CFCQL
[8]https://github.com/zbzhu99/madiff
[9]https://github.com/xmu-rl-3dv/DoF
[10]https://github.com/ikostrikov/implicit_q_learning

### D.4 PRETRAIN CRITIC MODELS

In this section, we provide a detailed explanation of the pre-training process for the critic networks. The network structures and parameter settings are consistent with those described in the previous section. We pre-trained two types of critic networks: independent critic networks and joint action learning critic networks. For the independent critic networks, each agent's input consists of the concatenation of its individual dataset's states and actions, with the network learning each agent's behavior independently. In contrast, the joint action learning critic network adopts a centralized approach, where the input comprises the concatenated joint states (observations) and joint actions of all agents, enabling a global perspective for joint decision-making. All pre-trained critics were trained for 200-500 epochs with checkpoints saved every 50 epochs. In subsequent OMSD training, the critic generally loads the checkpoint from the final epoch.

During the optimization process, we made adjustments to various hyperparameters and design choices, uncovering some important insights. First, the temperature and quantile regression coefficient $\tau$ were found to significantly affect the performance of pre-trained IQL. We performed a sweep of $\tau$ values in the range of [0.3, 0.5, 0.7, 0.9] and temperature values in the range of [1, 3, 5, 7, 10] across datasets of different quality and reported the optimal hyperparameters in Tables 6 and 7. Second, regarding the clamping of the advantage function, we initially clamped the exponential advantage term `exp_adv` at a maximum value of 100. However, we later tried directly restricting the advantage values to the range [-1, 1], which improved training stability in certain cases.

However, in the `MPE` environment, we encountered some challenges and issues that significantly impacted OMSD's performance. First, in medium replay datasets compared to those of other quality levels, the training speed was approximately 3 times faster than expected. Additionally, the resulting performance failed to learn meaningful signals. We hypothesize this is due to the sample volume of medium replay datasets being significantly lower than that of others, with medium replay containing only 62,500 samples, whereas datasets of other quality levels contain 1,000,000 samples. The poor performance may be influenced by the dataset's characteristics or overfitting during training, which requires further investigation and resolution. Notably, such issues were not observed in datasets from other environments, such as `MaMuJoCo`.

#### D.4.1 MPE

Since `MPE` tasks consist of only 25 steps per episode, significantly fewer than the 1000 steps per episode in MaMuJoCo, we follow the settings of Clean Offline RL Tarasov et al. (2023) to train IQL algorithms 500 epochs with 1000 update steps per epoch. Below are the hyperparameters for all three `MPE` tasks:

#### D.4.2 MAMUJOCO

The training parameters are aligned with SRPO and have been shown to work effectively. Specifically, for the critic, we use 10,000 steps per epoch for a total of 200 epochs. The quantile regression coefficient $\tau$ is set to 0.9 for maze tasks and 0.7 otherwise, while the temperature $\beta$ is fixed at 10. Additionally, the exponential advantage term "`exp_adv`" is clamped to a maximum value of 100 to ensure training stability.

For the MaMuJoCo tasks, the hyperparameters are outlined as follows. The dataset `2-HalfCheetah 200` is derived from OMAR, whereas the dataset `2-HalfCheetah 210` is sourced from OG-MARL Formanek et al. (2023) and MADiff Zhu et al. (2024).

### D.5 PRETRAIN DIFFUSION MODELS

For diffusion models, we follow the SRPO Chen et al. (2024) settings with slight modifications to improve training efficiency. Specifically, we reduce the number of layers from 3 to 2. The noise settings are defined as $t = \texttt{torch.rand}(a.\texttt{shape}[0], \texttt{device} = s.\texttt{device}) \times 0.96 + 0.02$. For the base SRPO framework, we use a hidden dimension of 64, a $\tau$ target network soft update rate of 0.01, a learning rate of 0.01, and the Annealing AdamW optimizer. Denoising is performed with 20 steps, while the denoising DDPM model operates with 5 steps using a beta schedule set to the "vp" strategy.

Table 6: IQL Training Hyperparameters in MPE

| Environment | Task | Hyper Parameter Name | Value |
|---|---|---|---|
| Global | | Training Steps/Epoch | 1000 |
| | | Epochs | 500 |
| Cooperative Navigation | Expert | temperature | 3.0 |
| | Expert | $\tau$ | 0.5 |
| | Medium | temperature | 0.5 |
| | Medium | $\tau$ | 0.7 |
| | Random | temperature | 0.5 |
| | Random | $\tau$ | 0.5 |
| Predator Prey | Expert | temperature | 7.0 |
| | Expert | $\tau$ | 0.7 |
| | Medium | temperature | 1.0 |
| | Medium | $\tau$ | 0.5 |
| | Random | temperature | 5.0 |
| | Random | $\tau$ | 0.7 |
| World | Expert | temperature | 3.0 |
| | Expert | $\tau$ | 0.5 |
| | Medium | temperature | 1.0 |
| | Medium | $\tau$ | 0.9 |
| | Random | temperature | 7.0 |
| | Random | $\tau$ | 0.7 |

Table 7: IQL Training Hyperparameters in `MaMuJoCo`

| Environment | Task | Hyper Parameter Name | Value |
|---|---|---|---|
| Global | | Training Steps/Epoch | 10000 |
| | | Epochs | 200 |
| 2-HalfCheetah 200 | Expert | temperature | 3.0 |
| | Expert | $\tau$ | 0.7 |
| | Medium | temperature | 3.0 |
| | Medium | $\tau$ | 0.7 |
| | Medium-Replay | temperature | 3.0 |
| | Medium-Replay | $\tau$ | 0.7 |
| | Random | temperature | 5.0 |
| | Random | $\tau$ | 0.5 |

In this study, we pretrained three types of diffusion models: (1) the independent diffusion model, (2) the joint action learning diffusion model, and (3) the sequential diffusion model. In the independent diffusion model, each agent's input consists of a concatenation of its individual dataset's state and action. For the joint action learning diffusion model, learning is treated as a centralized process, with inputs comprising the concatenated joint states (observations) and joint actions of all agents. Finally, the sequential diffusion model extends this idea by incorporating the preceding agents' actions as a prefix to the input. Combined with each agent's own state and action, this adjustment results in task-specific variations in input dimensionality for each agent. The hyperparameters are shown in Tables 8 and Table 9.

### D.5.1 MPE

Here are the hyperparameters for all three tasks in `MPE` environments shown in Table 8.

Table 8: Diffusion Models Training Hyperparameters in MPE

| Environment | Task | Hyper Parameter Name | Value |
|---|---|---|---|
| Global | | Training Steps | 100000 |
| | | Annealing Epochs | 10 |
| Cooperative Navigation | Expert | $\beta$ | 0.001 |
| | Medium | $\beta$ | 0.005 |
| | Random | $\beta$ | 0.05 |
| Predator Prey | Expert | $\beta$ | 0.005 |
| | Medium | $\beta$ | 0.05 |
| | Random | $\beta$ | 0.5 |
| World | Expert | $\beta$ | 0.01 |
| | Medium | $\beta$ | 0.05 |
| | Random | $\beta$ | 0.5 |

### D.5.2 MaMuJoCo

Here are the hyperparameters for `MaMuJoCo` comes from OMAR Pan et al. (2022) and MADiff Zhu et al. (2024) shown in Table 9.

Table 9: Diffusion Models Training Hyperparameters in MaMuJoCo

| Environment | Task | Hyper Parameter Name | Value |
|---|---|---|---|
| Global | | Traning Steps | 100000 |
| | | Annealing Epochs | 10 |
| 2-HalfCheetah 200 | Expert | $\beta$ | 0.001 |
| | Medium | $\beta$ | 0.005 |
| | Medium-Replay | $\beta$ | 0.05 |
| | Random | $\beta$ | 0.05 |

### D.6 TRAIN OMSD MODELS

In this subsection, we provide the hyperparameters for training OMSD models.

### D.6.1 MPE

Here are the hyperparameters for all three tasks in `MPE` environments as shown in Table 10.

### D.6.2 MaMuJoCo

Here are the hyperparameters for `MaMuJoCo`. The dataset 2-HalfCheetah 200 comes from OMAR (Pan et al., 2022), and the dataset 2-HalfCheetah 210 comes from MADiff (Zhu et al., 2024) as shown in Table 11.

### D.7 MORE ABLATION STUDY RESULTS

### D.7.1 SCORE DECOMPOSITION METHODS

Here we present more ablation study results of all three `MPE` tasks in Fig. 7, i.e., Cooperative Navigation, Predator Prey, and World. Over multiple quality datasets across various tasks, our methods demonstrates advantages over pre-trained Critic IQL and other policy decomposition methods.

Table 10: OMSD Training Hyperparameters in MPE

| Environment | Task | Hyper Parameter Name | Value |
|---|---|---|---|
| Global | | Training Steps | 100000 |
| | | Annealing Epochs | 10 |
| Cooperative Navigation | Expert | $\beta$ | 0.001 |
| | Medium | $\beta$ | 0.005 |
| | Random | $\beta$ | 0.05 |
| Predator Prey | Expert | $\beta$ | 0.005 |
| | Medium | $\beta$ | 0.05 |
| | Random | $\beta$ | 0.5 |
| World | Expert | $\beta$ | 0.01 |
| | Medium | $\beta$ | 0.05 |
| | Random | $\beta$ | 0.5 |

Table 11: OMSD Training Hyperparameters in MaMuJoCo

| Environment | Task | Hyper Parameters Name | Value |
|---|---|---|---|
| Global | | Traning Steps | 100000 |
| | | Annealing Epochs | 10 |
| 2-HalfCheetah 200 | Expert | $\beta$ | 0.001 |
| | Medium | $\beta$ | 0.005 |
| | Medium-Replay | $\beta$ | 0.05 |
| | Random | $\beta$ | 0.05 |

### D.7.2 HYPERPARAMS

For the temperature coefficient, we sweep over $\beta \in \{0.01, 0.02, 0.05, 0.1, 0.2, 0.5\}$ and observe large variances in appropriate values across different tasks (Fig. 8). We speculate this might be due to $\beta$ being closely intertwined with the behavior distribution and the variance of the Q-value. These factors might exhibit entirely different characteristics across diverse tasks. Our choices for $\beta$ are detailed in Table .

### D.7.3 VISUALIZATION OF FINAL POLICY

In Fig. 9, we illustrate the full learning trajectories of OMSD algorithms on `MPE` datasets.

The gray data points represent the t-SNE (Van der Maaten & Hinton, 2008) distribution of the state-joint action pairs from the original dataset, while the data points transitioning from light blue to dark blue indicate the t-SNE distribution of episode trajectories collected under policies at different training steps, using 10 random seeds. It can be observed that during the policy update process, the distribution remains mostly within the range of the original dataset, effectively avoiding the OOD problem. This demonstrates that our sequential score decomposition method can effectively ensure that the learning distribution remains in-sample under multimodal offline MARL datasets.

Furthermore, as the policy updates, the policy gradually learns and converges to high-reward regions, concentrating within a limited range. This indicates that the joint action critic can effectively provide signals for high-reward regions, guiding policy improvement.

### D.7.4 SEQUENTIAL UPDATE ORDERS

To demonstrate our method's insensitivity to update order, we conducted randomized ordering experiments on the OMIGA Hopper-v2 datasets. Specifically, the task involved three agents. The standard OMSD training process used the default agent ID order as the pre-trained diffusion model and policy update order to determine prefix actions (0-1-2). In addition, we randomly assigned update

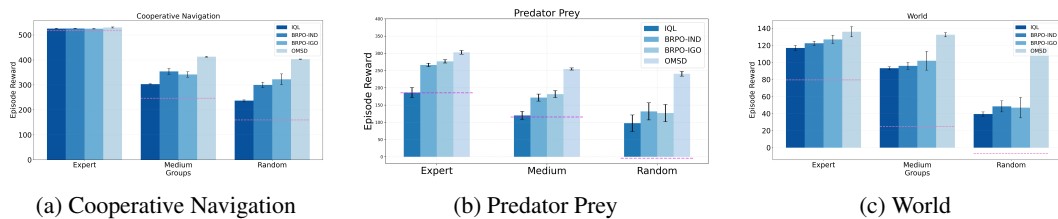

Figure 7: Comparasion of Pretrained IQL, BRPO-IND, BRPO-CTDE, and OMSD on Cooperative Navigation, Predator Prey, and World Tasks.

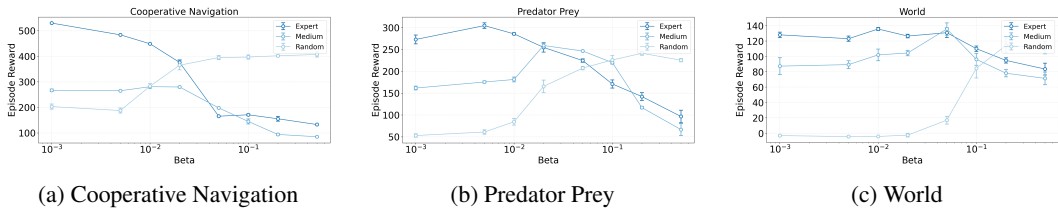

Figure 8: Comparison of regularization term $\beta$ of OMSD on Cooperative Navigation, Predator Prey, and World Tasks.

orders of 0-2-1 and 2-1-0 as control groups to avoid accidental agent relationship modeling under specific update orders. Experimental results show that, with the same pre-training parameters and OMSD training parameter settings, changing only the update order does not significantly impact performance, strongly demonstrating the robustness of our method for capturing complex multimodal behavior distributions. Furthermore, thanks to our structural design, our algorithm only needs to consider the behavior of preceding agents during training, relying solely on its own local observations during execution without needing action information from others. Compared to sequential action modeling methods such as MAT (Wen et al., 2022), this method offers greater flexibility and is insensitive to specific agent dependencies.

## E   DATA QUALITY VISUALIZATION OF OFFLINE DATASETS

In this section, we provide more details about the offline datasets `MPE`, 2-agent `HalfCheetah` we used in this paper. The data distribution with violin plots and histogram plots in Fig. 12, Fig. 11, and Fig. 13. These plots are provided by OG-MARL[11] Formanek et al. (2023).

## F   WHY DO OFFLINE INDEPENDENT LEARNING AND NAIVE CTDE FRAMEWORKS FAIL?

To further elucidate the impact of multimodal behavioral policies on offline MARL, we selected the standard policy-based offline RL method, BRPO Wu et al. (2019), and extended it to the MARL setting to analyze the failure modes. We focused on two mainstream paradigms: independent learning and CTDE learning.

### F.1   POLICY-BASED OFFLINE MARL WITH INDEPENDENT LEARNING.

We begin our analysis with independent BRPO (BRPO-IND), a fundamental case under the independent learning paradigm. Generally, independent learning methods decompose MARL problems into multiple autonomous single-agent RL processes by treating other agents as part of dynamic environments. This is a robust approach widely adopted in both online and offline MARL algorithms that has demonstrated stable performance across many tasks, which assumes that each policy is independently factorizable. Specifically, in BRPO-IND, each agent independently learns the critic and

---

[11]https://github.com/instadeepai/og-marl

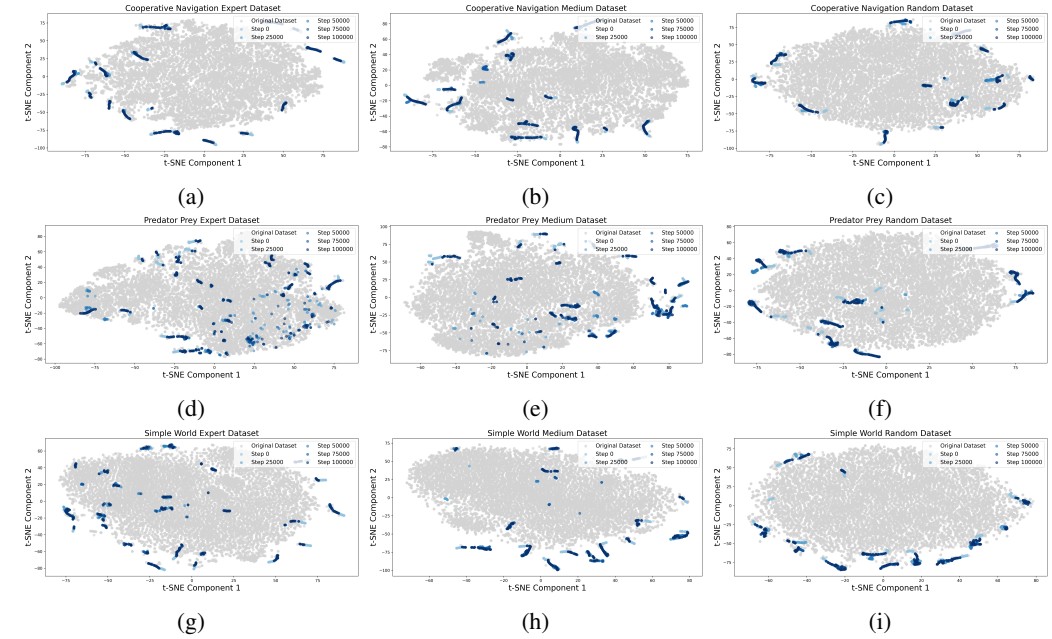

Figure 9: Full training trajectories of OMSD on `MPE` tasks.

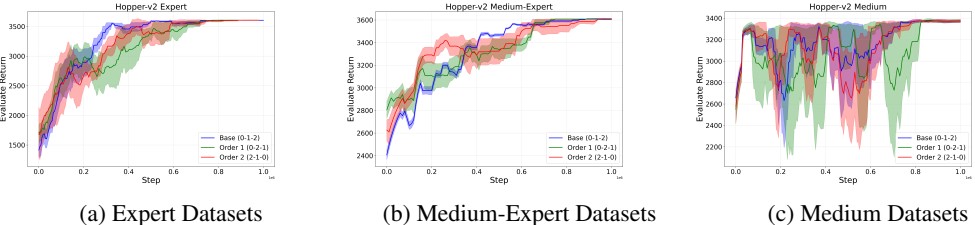

(a) Expert Datasets  (b) Medium-Expert Datasets  (c) Medium Datasets

Figure 10: Ablation experiments on three different random update orders of agents in Hopper-v2.

models individual behavior policy $\mu_i(a_i|s)$ from individual datasets. With Lemma 2.1, we propose the following proposition.

**Proposition F.1.** *Consider a fully cooperative game with $n$ agents. Under the independent learning framework, the optimal individual policy of each agent is:*

$$\pi_i^*(a_i \mid s) = \frac{1}{Z(s)}\mu_i(a_i \mid s)\exp\left(\beta_i Q^i(s, a_i)\right),$$

*where $\mu_i$ and $Q^i$ are individual behavior policy and Q-value function of agent $i$, respectively. With Lemma 2.1, the learning objective of BRPO-IND is:*

$$\mathcal{L}_{Ind} = \max \sum_{i=1}^{n} \mathbb{E}_{s\sim\mathcal{D}_\mu, a_i\sim\pi_{\theta_i}} Q^i(s, a_i) - \underbrace{\frac{1}{\beta} D_{KL}\left[\pi_{\theta_i}\|\mu_i\right]}_{Ind\ Behavior\ Reg}.$$

Here, the KL penalty prevents the learned individual policy from diverging significantly from the individual behavior policy. By taking the gradient of equation $\mathcal{L}_{Ind}$ with respect to each agent's policy parameters, we obtain:

$$\nabla_{\theta_i}\mathcal{L}_{Ind} = \mathbb{E}_{s\sim\mathcal{D}^\mu}\left[\nabla_{a_i}Q^i(s, a_i)\big|_{a_i=\pi_{\theta_i}} + \frac{1}{\beta}\underbrace{\nabla_{a_i}\log\mu_i(a_i \mid s)\big|_{a_i=\pi_{\theta_i}(s)}}_{=-\epsilon_i^*(a_t|s,t)/\sigma_t|_{t\to0}}\right]\nabla_{\theta_i}\pi_{\theta_i}(a_i|s), \quad (7)$$

where $\epsilon_i^*(a_t \mid s, t)$ represents the score function of individual behavior policy $\nabla_{a_i}\mu_i(a_i|s)$ (Song et al., 2020a).

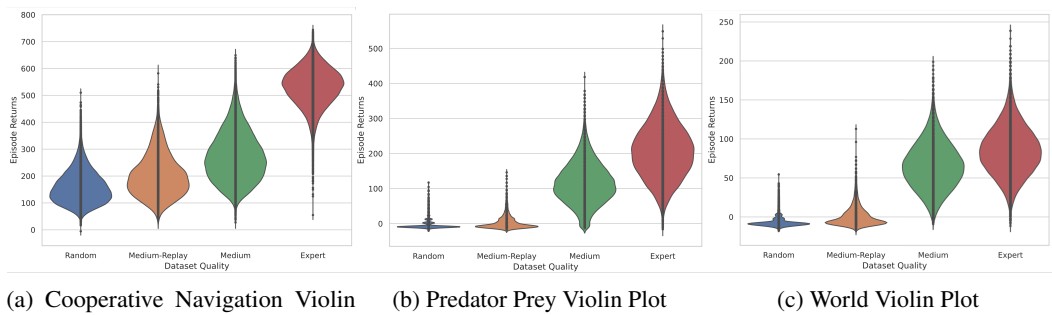

(a) Cooperative Navigation Violin Plot

(b) Predator Prey Violin Plot

(c) World Violin Plot

Figure 11: Violin plots of MPE offline datasets.

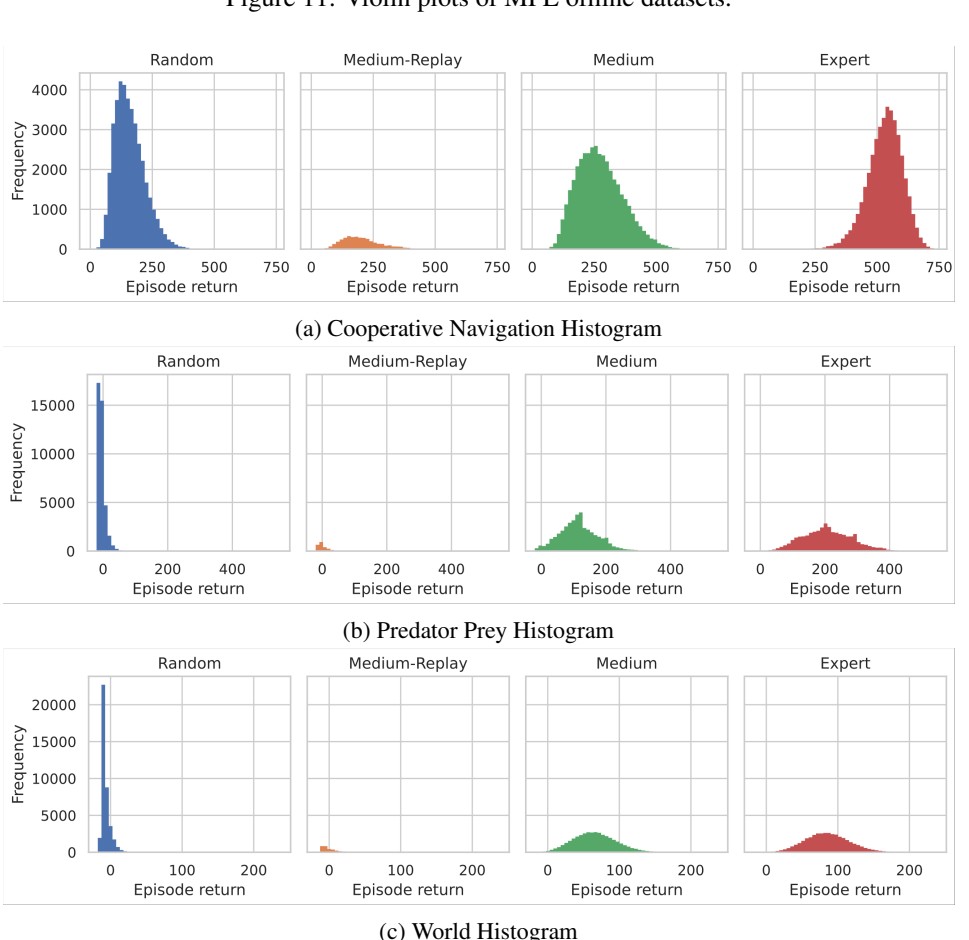

(a) Cooperative Navigation Histogram

(b) Predator Prey Histogram

(c) World Histogram

Figure 12: Histogram plots of MPE offline datasets.

### F.2 POLICY-BASED OFFLINE MARL WITH CTDE LEARNING.

In the CTDE framework, the centralized training process typically leverages the actions of other agents, global states, and the policies of other agents to learn the optimal joint policy. It can stabilize nonstationary learning process by capture interactive relationships between agents and global information. The executable individual policies are ususally distilled through value decomposition or policy decomposition. In policy-based methods, such as FOP (Zhang et al., 2021b) and AlberDICE (Matsunaga et al., 2023), the decomposable assumption IGO (Individual-Global-Optimal) $\pi_\Psi^* :=$ $\pi_{\psi^i}^{i*} \prod_{j=-i} \pi_{\psi^j}^{j*}$ is typically used to extract individual policies from the joint optimal policy. Based on IGO principle and Lemma 2.1, we propose the BRPO-CTDE as follows.

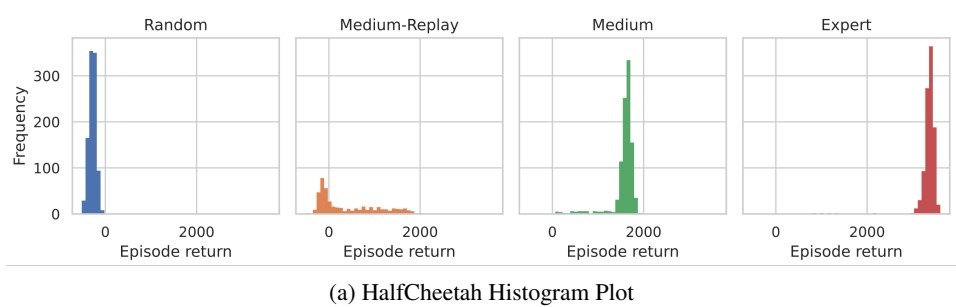

(a) HalfCheetah Histogram Plot

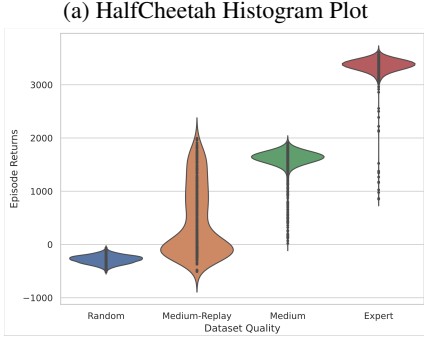

(b) HalfCheetah Violin Plot

Figure 13: Histogram and Violin plots of MaMuJoCo offline datasets.

**Proposition F.2.** *Consider a fully cooperative game with n agents. In centralized learning process, the optimal joint policy is derived as*

$$\pi^*(\boldsymbol{a} \mid s) = \frac{1}{Z(s)} \mu(\boldsymbol{a} \mid s) \exp\left(\beta Q^{tot}(s, \boldsymbol{a})\right),$$

*where $\boldsymbol{a}$ represents the joint actions and $Q^{tot}$ represents the global state-action value function. With Lemma 2.1 and the factorization principle, the learning objective for each agent becomes*

$$\mathcal{L}_{CTDE}^i = \min_{\theta_i} \mathbb{E}_{s \sim \mathcal{D}_\mu} D_{\mathrm{KL}}[\pi_{\theta^i}(\cdot \mid s)\pi_{\theta^{-i}}(\cdot \mid s) \| \pi^*(\cdot \mid s)]$$

$$= \max_{\theta_i} \mathbb{E}_{s \sim \mathcal{D}^\mu, \boldsymbol{a} \sim \pi_\theta(\cdot | s)} Q^{tot}(s, \boldsymbol{a}) - \frac{1}{\beta} \underbrace{D_{\mathrm{KL}}\left[\pi_{\theta^i}(\cdot \mid s)\pi_{\theta^{-i}}(\cdot \mid s) \| \mu(\boldsymbol{a} \mid s)\right]}_{\textit{Joint Behavior Reg}}.$$

Compared to BRPO-IND, BRPO-CTDE minimizes the KL divergence between the learned joint policy $\Pi_i^n \pi_i(a_i|s)$ and the joint behavior policy distribution $\mu(\boldsymbol{a}|s)$ on each agent's policy update. Then we can derive the gradient of equation $\mathcal{L}_{CTDE}^i$ with respect to each agent's policy parameters as:

$$\nabla_{\theta_i} \mathcal{L}_{CTDE}^i = \mathbb{E}_{s \sim \mathcal{D}^\mu, a^{-i} \sim \pi_{\theta_{-i}}} \left[ \nabla_{a_i} Q^{tot}(s, \boldsymbol{a})\big|_{a = \pi_\theta(\cdot|\boldsymbol{s})} + \frac{1}{\beta} \nabla_{a_i} \log \mu(\boldsymbol{a} \mid s)\big|_{a_i = \pi_{\theta_i}(s)} \right] \nabla_{\theta_i} \pi_{\theta_i}(a_i|s).$$
$$(8)$$

Equations (7) and (8) reveal that the gradients in offline policy-based MARL consist of Q-value gradients and behavior policy regularization terms. However, this structure poses significant challenges for joint policy updates.

First, an obvious problem arises in the coordination of Q-value gradients. In offline MARL, the absence of online data collection severely limits the ability to adjust policies by exploring new experiences. This issue further exacerbates the misalignment coordination of individual Q-value gradients in MARL and may lead to suboptimal gradient directions (Kuba et al., 2022; Pan et al., 2022).

Admittedly, the CTDE frameworks can slightly alleviate the Q-value gradients coordination problem by directly providing local gradients of the joint Q-function to each agent. However, the individual

regularization terms are also challenging due to the multi-modal property of the joint behavior policy $\mu(\boldsymbol{a}|s)$. With IGO assumption, the individual behavior regularization term in CTDE becomes a biased score function as

$$\nabla_{a_i} \log \mu(a \mid s) = \nabla_{a_i} \pi(a|s) \nabla_a \log \mu(\boldsymbol{a} \mid s)$$
$$\neq \nabla_{a_i} \log \mu(a^i \mid s),$$

where $\nabla_\pi \log \mu(a|s)$ represents the score function of the joint behavior policy captured by high-capacity generative models, and $\nabla_{a_i} \pi$ is the partial gradient of the joint policy with respect to agent $i$. The primary difficulty lies in accurately calculating $\nabla_{a_i} \pi$ from the multi-modal joint behavior policy, as the offline joint policy may not be easily factorizable into individual agent policies.

These challenges faced by BRPO-IND and BRPO-CTDE are fundamentally rooted in the multi-modality problem described in Section 3.1 and can be generalized to other policy-based offline RL algorithms. Multi-modal joint behavior policies cause complex dependencies among agents, while the infactorization property prevents accurate factorization of these joint policies. Directly applying assumptions in online MARL, such as the factorization assumption, will induce biased policy regularization on individual policy update, ultimately causing the joint policy distribution to deviate from the support set of the dataset.

## G  THEOREM DETAILS

### G.1  PROOF OF PROPOSITION 3.1

We consider a fully-cooperative n-player game with a single state and action space $A = [0, 1]^n$. Let $\pi^*$ be the optimal joint policy with two optimal modes: $a_1 = (1, \ldots, 1)$ and $a_2 = (0, \ldots, 0)$. Let $\hat{\pi}$ be a factorized approximation of $\pi^*$ such that $\hat{\pi}(a) = \prod_{i=1}^n \hat{\pi}_i(a_i)$, where each $\hat{\pi}_i$ is learned independently.

Given that $\pi^*$ has two optimal modes $(1, \ldots, 1)$ and $(0, \ldots, 0)$, and each $\hat{\pi}_i$ is learned independently, the best approximation for each individual policy is to assign equal probability to 0 and 1. Thus, each $\hat{\pi}_i$ converges to Uniform$(\{0, 1\})$, with $\hat{\pi}_i(0) = \hat{\pi}_i(1) = 0.5$ for all $i$.

Since each $\hat{\pi}_i$ is Uniform$(\{0, 1\})$, the joint policy $\hat{\pi}$ will have a mode for each possible combination of 0s and 1s across the $n$ players. There are $2^n$ such combinations. The probability of each mode is $\hat{\pi}(a) = \prod_{i=1}^n \hat{\pi}_i(a_i) = (0.5)^n = 2^{-n}$. Therefore, the reconstruction of joint policy $\hat{\pi}$ exhibits $2^n$ modes, each with probability $2^{-n}$.

To prove that the total variation distance between $\pi^*$ and $\hat{\pi}$ is $\delta_{TV}(\pi^*, \hat{\pi}) = 1 - 2^{1-n}$, we start with the definition of total variation distance:

$$\delta_{TV}(\pi^*, \hat{\pi}) = \frac{1}{2} \sum_a |\pi^*(a) - \hat{\pi}(a)|$$

For $\pi^*$, we have $\pi^*(a_1) = \pi^*((1, \ldots, 1)) = 0.5$, $\pi^*(a_2) = \pi^*((0, \ldots, 0)) = 0.5$, and $\pi^*(a) = 0$ for all other $a$. For $\hat{\pi}$, we have $\hat{\pi}(a) = 2^{-n}$ for all $2^n$ modes.

Calculating the sum of absolute differences:

$$|\pi^*(a_1) - \hat{\pi}(a_1)| + |\pi^*(a_2) - \hat{\pi}(a_2)| = |0.5 - 2^{-n}| + |0.5 - 2^{-n}| = 1 - 2^{1-n}$$

For the remaining $2^n - 2$ modes of $\hat{\pi}$:

$$\sum |0 - 2^{-n}| = (2^n - 2) \cdot 2^{-n} = 1 - 2^{1-n}$$

Therefore,

$$\delta_{TV}(\pi^*, \hat{\pi}) = \frac{1}{2} \cdot (1 - 2^{1-n} + 1 - 2^{1-n}) = 1 - 2^{1-n}$$

As $n \to \infty$, we have:

$$\lim_{n \to \infty} \delta_{TV}(\pi^*, \hat{\pi}) = \lim_{n \to \infty} (1 - 2^{1-n}) = 1 - \lim_{n \to \infty} 2^{1-n} = 1 - 0 = 1$$

This limit indicates a severe distribution shift between the true optimal policy $\pi^*$ and its factorized approximation $\hat{\pi}$ as the number of players increases.

## G.2  PROOF OF PROPOSITION F.1

First, we derive the optimization objectives with independent learning framework. By decomposing the KL term in (F.1), we have

$$\mathcal{L}_{Ind} = \sum_{i=1}^{n} \left( \mathbb{E}_{s \sim \mathcal{D}_\mu, a_i \sim \pi_{\theta_i}} Q^i(s, a_i) + \frac{1}{\beta} \mathbb{E}_{s \sim \mathcal{D}^\mu, a_i \sim \pi_{\theta_i}} \log \mu_i(a_i|s) + \frac{1}{\beta} \mathbb{E}_{s \sim \mathcal{D}^\mu} \mathcal{H}(\pi_i(a_i|s)) \right)$$

where $\mathcal{H}(\pi_i(a_i|s))$ is the entropy of the agent $i$'s policy. As BRPO-IND learns behavior policy independently, we can directly get the term $\log \mu_i(a_i|s)$ implicitly from the pretrained diffusion models of each agent.

Consider that each agent's policy is trained independently without dependency, we can derive the gradient of agent $i$ as

$$\nabla_{\theta_i} \mathcal{L}_{Ind} = \nabla_{\theta_i} \sum_{i=1}^{n} \left( \mathbb{E}_{s \sim \mathcal{D}_\mu, a_i \sim \pi_{\theta_i}} Q^i(s, a_i) + \frac{1}{\beta} \mathbb{E}_{s \sim \mathcal{D}^\mu, a_i \sim \pi_{\theta_i}} \log \mu_i(a_i|s) + \frac{1}{\beta} \mathbb{E}_{s \sim \mathcal{D}^\mu} \mathcal{H}(\pi_i(a_i|s)) \right)$$

$$= \mathbb{E}_{s \sim \mathcal{D}_\mu, a_i \sim \pi_{\theta_i}} \left[ \nabla_{\theta_i} Q^i(s, a_i) + \frac{1}{\beta} \nabla_{\theta_i} \log \mu_i(a_i|s) \right]$$

$$= \mathbb{E}_{s \sim \mathcal{D}^\mu, a_i \sim \pi_{\theta_i}} \left[ \nabla_{\theta_i} \pi_i * \nabla_{a_i} Q^i(s, a_i) + \frac{1}{\beta} \nabla_{\theta_i} \pi_i * \nabla_{a_i} \log \mu_i(a_i|s) \right]$$

$$= \mathbb{E}_{s \sim \mathcal{D}^\mu, a_i \sim \pi_{\theta_i}} \left[ \nabla_{a_i} Q^i(s, a_i) + \frac{1}{\beta} \nabla_{a_i} \log \mu_i(a_i|s) \right] \nabla_{\theta_i} \pi_i.$$

Notice that the term $\nabla_{a_i} \log \mu_i(a_i|s)$ serves as the score function of the independent behavior policy, we can further construct a surrogate loss $\mathcal{L}_{Ind}^{surr}$ and derive a practical gradient for BRPO-IND. Our proof is mainly inspired by the following Lemma G.1.

**Lemma G.1** (Proposition 1 in Chen et al. (2024)). *Given that $\pi$ is sufficiently expressive, for any time $t$, any state $s$, we have*

$$\arg \min_\pi D_{KL}[\pi_t(\cdot|s)\|\mu_t(\cdot|s)] = \arg \min_\pi D_{KL}[\pi(\cdot|s)\|\mu(\cdot|s)],$$

*where both $\mu_t$ and $\pi_t$ follow the same predefined diffusion process in $q_{t_0}(x_t|x_0) = \mathcal{N}(x_t|\alpha_t x_0, \sigma_t^2 I)$, which implies $x_t = \alpha_t x_0 + \sigma_t \varepsilon$.*

The surrogate loss is

$$L_{Ind}^{surr}(\theta_i) = \mathbb{E}_{s, a_i \sim \pi_{\theta_i}} Q(s, a_i) - \frac{1}{\beta} \mathbb{E}_{t,s} \omega(t) \frac{\sigma_t}{\alpha_t} D_{KL}[\pi_{\theta_i, t}(\cdot|s)\|\mu_{i,t}(\cdot|s)]. \tag{9}$$

Then we can propose the practical gradient as follows.

**Proposition G.2** (Practical Gradient of BRPO-IND). *Given that $\pi_{\theta_i}$ is deterministic policy and $\epsilon_i^*$ is the optimal diffusion model of independent behavior policy $\mu_i$, the gradient of the surrogate loss (9) w.r.t agent $i$ is*

$$\nabla_{\theta_i} L_{surr}^\pi(\theta) = \left[ \mathbb{E}_s \nabla_a Q_\phi(s, a)|_{a=\pi_\theta(s)} - \frac{1}{\beta} \mathbb{E}_{t,s} \omega(t)(\epsilon_i^*(a_{t,i}|s, t) - \epsilon_i)|_{a_{i,t} = \alpha_t \pi_{\theta_i}(s) + \sigma_t \epsilon_i} \right] \nabla_{\theta_i} \pi_{\theta_i}(s).$$

*Proof.* The fundamental framework of the proof follows the proof process of SRPO (Chen et al., 2024), extending it to the multi-agent scenario. Based on the forward diffusion process in section 2.2, we can represent the noisy distribution of actor policy at step $t$ as

$$\pi_{\theta_i, t}(a_{t,i}|s) = \int \mathcal{N}(a_{i,t}|\alpha_t a_i, \sigma_t^2 I) \pi_{\theta_i}(a_i|s) da_i \tag{10}$$

$$= \int \mathcal{N}(a_{t,i}|\alpha_t a_i, \sigma_t^2 I) \delta(a_i - \pi_{\theta_i}(s)) da_i \tag{11}$$

$$= \mathcal{N}(a_{t,i}|\alpha_t \pi_{\theta_i}(s), \sigma_t^2 I) \tag{12}$$

Note that $\pi_{\theta,t}(\cdot|s)$ is a Gaussian policy with expected value $\alpha_t\pi_\theta(s)$ and variance $\sigma_t^2 I$, we can simplify the surrogate training objective as

$$
L_{Ind}^{surr}(\theta_i) = \mathbb{E}_{s,a_i\sim\pi_{\theta_i}(\cdot|s)}Q(s,a_i) - \frac{1}{\beta}\mathbb{E}_{t,s}\omega(t)\frac{\sigma_t}{\alpha_t}D_{\mathrm{KL}}[\pi_{\theta_i,t}(\cdot|s)\|\mu_{i,t}(\cdot|s)]
$$

$$
= \mathbb{E}_s Q(s,a_i)|_{a_i=\pi_{\theta_i}(s)} + \frac{1}{\beta}\mathbb{E}_{t,s}\omega(t)\frac{\sigma_t}{\alpha_t}\mathbb{E}_{a_{i,t}\sim\mathcal{N}(\cdot|\alpha_t\pi_{\theta_i}(s),\sigma_t^2 I)}[\log\mu_t(a_{i,t}|s) - \log\pi_{t,\theta_i}(a_{i,t}|s)]
$$

Then we can derive the gradient of this objective as follows

$$
\nabla_{\theta_i}\mathcal{L}_{Ind}^{surr}(\theta_i) = \nabla_{\theta_i}\mathbb{E}_{\boldsymbol{s}\sim\mathcal{D}^\mu}Q_\phi(\boldsymbol{s},a_i)|_{a_i\sim\pi_\theta^i(\boldsymbol{s})} + \frac{1}{\beta}\mathbb{E}_{t,s}\frac{\sigma_t}{\alpha_t}\omega(t)\nabla_{\theta_i}\mathbb{E}_{\epsilon_i}\left[\log\mu_t^i(a_t^i|s) - \log\pi_t^i(a_t^i|s)\right]
$$

$$
\text{(reparameterization of } \pi_i = \alpha_t\pi_{\theta_i}(s) + \sigma_t\epsilon_i)
$$

$$
= \nabla_{\theta_i}\mathbb{E}_{\boldsymbol{s}\sim\mathcal{D}^\mu}Q_\phi(\boldsymbol{s},a_i)|_{a_i\sim\pi_\theta^i(\boldsymbol{s})} + \frac{1}{\beta}\mathbb{E}_{t,s,\epsilon_i}\frac{\sigma_t}{\alpha_t}\omega(t)\left[\nabla_{\theta_i}\log\mu_t^i(a_t^i|s) - \nabla_{\theta_i}\log\pi_t^i(a_t^i|s)\right] \quad \text{(chain rule)}
$$

$$
= \nabla_{\theta_i}\mathbb{E}_{\boldsymbol{s}\sim\mathcal{D}^\mu}Q_\phi(\boldsymbol{s},a_i)|_{a_i\sim\pi_\theta^i(\boldsymbol{s})} + \frac{1}{\beta}\mathbb{E}_{t,s,\epsilon_i}\frac{\sigma_t}{\alpha_t}\omega(t)\left[\nabla_{a_i^t}\log\mu_t^i(a_t^i|s)\nabla_{\theta_i}a_i^t|_{a_i^t=\alpha_t\pi_{\theta_i}(s)+\sigma_t\epsilon_i}\right.
$$

$$
\left. - \nabla_{a_i^t}\log\pi_t^i(a_t^i|s)\nabla_{\theta_i}a_i^t|_{a_i^t=\alpha_t\pi_{\theta_i}(s)+\sigma_t\epsilon_i}\right]
$$

$$
= \mathbb{E}_{\boldsymbol{s}\sim\mathcal{D}^\mu}\nabla_{a_i}Q_\phi(\boldsymbol{s},\boldsymbol{a}_i,\boldsymbol{a}_{-i})|_{\boldsymbol{a}_i\sim\pi_\theta^i(\boldsymbol{s}),\boldsymbol{a}_{-i}\sim\pi_\theta^{-i}(\boldsymbol{s})}\nabla_{\theta_i}\pi_i
$$

$$
+ \frac{1}{\beta}\mathbb{E}_{t,s,\epsilon_i}\frac{\sigma_t}{\alpha_t}\omega(t)\left[-\frac{\epsilon_i(a_i|s,t)}{\sigma_t}\alpha_t\nabla_{\theta_i}\pi_{\theta_i}(s) + \frac{\epsilon}{\sigma_t}\alpha_t\nabla_{\theta_i}\pi_{\theta_i}(s)\right]
$$

$$
= \left[\underbrace{\mathbb{E}_{\boldsymbol{s}}\nabla_{a_i}Q_\phi(\boldsymbol{s},\boldsymbol{a}_i,\boldsymbol{a}_{-i})|_{\boldsymbol{a}_i\sim\pi_\theta^i(\boldsymbol{s}),\boldsymbol{a}_{-i}\sim\pi_\theta^{-i}(\boldsymbol{s})}}_{\text{Q gradient}}\right.
$$

$$
\left. -\frac{1}{\beta}\mathbb{E}_{t,s,\epsilon_i}\omega(t)\left(\underbrace{\epsilon_i(a_i^t|s,t)}_{\text{score }\mu_i^t} - \underbrace{\epsilon}_{\text{score }\pi_i^t}\right)|_{a_i^t=\alpha_t\pi_{\theta_i}(s)+\sigma_t\epsilon_i}\right]\nabla_{\theta_i}\pi_i(s)
$$

$$
\tag{13}
$$

$\square$

### G.3 Proof of Proposition F.2

First, we derive the optimization objectives with centralized learning framework. By decomposing the KL term, we have

$$
\mathcal{L}_{CTDE}^i = \mathbb{E}_{s\sim\mathcal{D}^\mu,\boldsymbol{a}\sim\pi_\theta(\cdot|s)}Q^{tot}(s,\boldsymbol{a}) + \frac{1}{\beta}\mathbb{E}_{s\sim\mathcal{D}^\mu,\boldsymbol{a}\sim\pi_\theta(\cdot|s)}\log\mu(\boldsymbol{a}|s) + \frac{1}{\beta}\mathbb{E}_{s\sim\mathcal{D}^\mu}\mathcal{H}(\pi(\boldsymbol{a}|s)),
$$

where $\mathcal{H}(\pi(\boldsymbol{a}|s))$ is the entropy of the joint policy. Then we need to distill the decentralized executive policy for each agent. Consider that each agent policy $\pi_{\theta_i}$ is an isotropic Gaussian policy, we can decompose the joint policy by $\pi = \pi_{\theta_i}\pi_{\theta_{-i}}$. The gradient of agent $i$ is as follows

$$
\nabla_{\theta_i}\mathcal{L}_{CTDE}^i = \nabla_{\theta_i}\mathbb{E}_{s\sim\mathcal{D}^\mu,\boldsymbol{a}_{-i}\sim\pi_{\theta_{-i}}(\cdot|s)}\left[Q^{tot}(s,\boldsymbol{a}) + \frac{1}{\beta}\log\mu(\boldsymbol{a}|s)\right] \tag{14}
$$

$$
= \mathbb{E}_{s\sim\mathcal{D}^\mu,\boldsymbol{a}_{-i}\sim\pi_{\theta_{-i}}(\cdot|s)}\left[\nabla_{\theta_i}Q^{tot}(s,\boldsymbol{a}) + \frac{1}{\beta}\nabla_{\theta_i}\log\mu(\boldsymbol{a}|s)\right] \tag{15}
$$

$$
= \mathbb{E}_{s\sim\mathcal{D}^\mu,\boldsymbol{a}_{-i}\sim\pi_{\theta_{-i}}(\cdot|s)}\left[\nabla_{\theta_i}\pi_i * \nabla_{a_i}Q^{tot}(s,\boldsymbol{a}) + \frac{1}{\beta}\nabla_{\theta_i}\pi_i * \nabla_{a_i}\log\mu(\boldsymbol{a}|s)\right] \tag{16}
$$

$$
= \mathbb{E}_{s\sim\mathcal{D}^\mu,\boldsymbol{a}_{-i}\sim\pi_{\theta_{-i}}(\cdot|s)}\left[\nabla_{a_i}Q^{tot}(s,\boldsymbol{a}) + \frac{1}{\beta}\nabla_{a_i}\log\mu(\boldsymbol{a}|s)\right]\nabla_{\theta_i}\pi_i. \tag{17}
$$

Importantly, different from the cases in BRPO-IND, we cannot distill a score function $\nabla_{a_i} \log \mu(\boldsymbol{a}|s)$ from the pretrained diffusion models of joint behavior policies. To illustrate the influence of inproporate factorizations, we slightly abuse the factorization assumptions to decompose the joint behavior policy as $\mu(\boldsymbol{a}|s) = \prod_{i=1}^{n} \mu_i(a_i|s)$ and propose a revised baseline called BRPO-CTDE. This variant shares most of the framework with BRPO-CTDE, but differs in the policy regularization component: instead of using the joint behavior policy, BRPO-CTDE employs individual behavior policies for regularization.

# H  DETAILS ABOUT PRACTICAL ALGORITHM

## H.1  OMSD PIPELINE

The OMSD methods contain a two-stages training process: 1) pretraining sequential diffusion models and joint action critic on the dataset by making score decomposition, and 2) injecting decomposed scores as the individual policy regularization terms into the critic and derive deterministic policies for execution. The resulting OMSD algorithm is presented in Algorithm 1.

The basic workflow of OMSD follows the idea of SRPO (Chen et al., 2024) by extending the single agent learning process into multi-agent process, where the unbiased score decomposition methods proposed in section 3.2 are plugged-in to avoid the uncoordination policy updated. Specifically, as we take the joint critic and individual score regularization, all the agents share the copies of a pre-trained common joint action Q-networks $Q_{tot}$ and keep individual pre-trained behavior diffusion models to extract the score regularization. This is a common setup in multi-agent reinforcement learning, such as MADDPG. Besides, each agent maintains a deterministic policy as the actor network, which bypasses the heavy iterative denoising process of diffusion models to generate actions and enjoy the fast decision-making speed.

## H.2  PRETRAINING IQL AS CRITIC

The centralized Q-network are pretrained with implicit Q-learning (Kostrikov et al., 2021), which introduced the expectile regression in pessimistic value estimation:

$$\min L_V(\zeta) = \mathbb{E}_{(s,a)\sim\mathcal{D}_\mu} \left[ L_2^\tau \left( Q_\phi(s,a) - V_\zeta(s) \right) \right],$$

$$\min L_Q(\phi) = \mathbb{E}_{(s,a,s')\sim\mathcal{D}_\mu} \left[ ||r(s,a) + \gamma V_\zeta(s') - Q_\phi(s,a)||_2^2 \right],$$

where $L_2^\tau(u) = |\tau - \mathbf{1}(u < 0)|u^2$ is the expectile operator.

## H.3  PRETRAINING DIFFUSION MODELS

Considering the state and actions are continuous, the behavior models are trained with classssifier-free guidance diffusion models (Hansen-Estruch et al., 2023; Chen et al., 2024) by minimizing the following loss:

$$\min_{\psi_i} L_\mu(\psi_i) = \mathbb{E}_{t,\epsilon_i,(s,a)\sim\mathcal{D}_\mu} \left[ ||\hat{\epsilon}_{\psi_i}(a_t^i|s, a^{i-}, t) - \epsilon||_2^2 \right]_{a_t^i = \alpha_t a^i + \sigma_t \epsilon}, \tag{18}$$

where $t \sim \mathcal{U}(0,1), \epsilon \sim \mathcal{N}(0,\mathbf{I})$, and the sequential score function can be estimated with $\hat{\epsilon}_{\psi_i}(a_t^i|s, a^{i-}, t) \approx -\sigma_t \nabla_{a_i} \log \mu(a_i|s, a^{i-})$ (Song et al., 2020a).

Following similar numerical computation simplification methods in SRPO Chen et al. (2024), we also utilize the intermediate distributions of the entire diffusion process $t \in [0,1]$ to replace the original training objective here. The surrogate objective is

$$\max_{\theta_i} \mathcal{L}_\pi^{surr}(\theta_i) = \mathbb{E}_{\boldsymbol{s}\sim\mathcal{D}^\mu, \boldsymbol{a}_i\sim\pi_i(\cdot|s), \boldsymbol{a}_{-i}\sim\pi_{-i}(\cdot|s)} Q_\phi(\boldsymbol{s}, \boldsymbol{a}_i, \boldsymbol{a}_{-i}) \tag{19}$$

$$- \frac{1}{\beta} \mathbb{E}_{t,s} \omega(t) \frac{\sigma_t}{\alpha_t} D_{\mathrm{KL}} \left[ \pi_{i,t}(\cdot \mid \boldsymbol{s}) \| \mu_{i,t}(\cdot \mid \boldsymbol{s}, a^{i-}) \right] |_{a^{i-}\sim\pi^{i-}},$$

where $\omega(t) = \delta(t - 0.02) \frac{\alpha_{0.02}}{\sigma_{0.02}}$ is the weighting parameters to ensure the gap between $\mathcal{L}^{surr}(\theta_i)$ and $\mathcal{L}(\theta_i)$, $\pi_{i,t}(\cdot \mid \boldsymbol{s}) := \mathbb{E}_{a_i\sim\pi_i(\cdot|s)}\mathcal{N}(a_{i,t}|\alpha_t a_i, \sigma_t^2 \mathbf{I})$, and $\mu_{i,t}(\cdot \mid \boldsymbol{s}, a^{i-}) := \mathbb{E}_{a_i\sim\mu_{i,t}(\cdot|\boldsymbol{s},a^{i-})}\mathcal{N}(a_{i,t}|\alpha_t a_i, \sigma_t^2 \mathbf{I})$.

Considering the instability of the diffusion model near the initial and terminal times, we truncate the time range as $t \sim \mathcal{U}(0.02, 0.98)$. Therefore, we can derive the practical gradients for optimizing the objective as

$$\nabla_{\theta_i} \mathcal{L}_\pi(\theta_i) = \mathbb{E}_{\boldsymbol{s} \sim \mathcal{D}^\mu, a^{i-} \sim \bar{\pi}^{-i}, a^{i+} \sim \pi^{i+}} \left[ \nabla_{\boldsymbol{a_i}} Q_\phi(\boldsymbol{s}, \boldsymbol{a}) \big|_{a_i = \pi_{\theta_i}, a_{-i} = \pi_{\theta_{-i}}(\boldsymbol{s})} \right. \tag{20}$$

$$+ \frac{1}{\beta} \underbrace{\nabla_{\boldsymbol{a_i}} \boldsymbol{a} \cdot \nabla_{\boldsymbol{a}} \log \mu(\boldsymbol{a} \mid \boldsymbol{s}) \big|_{\boldsymbol{a} = \boldsymbol{\pi}_\theta(\boldsymbol{s})}}_{= -\boldsymbol{\epsilon}^*(\boldsymbol{a_t}|\boldsymbol{s}, t)/\sigma_t|_{t \to 0}} \left. \nabla_{\theta_i} \pi_{\theta_i}(\boldsymbol{s}). \right.$$

Compared to the naive score decomposition methods BRPO-CTDE, the main improvement is replacing the biased score regularization with sequential decomposed score. It strongly guarantees the policy update directions and coordination among all agents' gradients.

### H.4 DISCUSSIONS

In OMSD, the sequential conditional distribution is solely utilized during the policy update phase to extract conditional score functions for policy regularization. Specifically, the sequential structure is not embedded in the execution policy. Instead, it is only used to model the joint behavior policy and derive score functions that guide individual policy updates. This design ensures that during execution, each agent's policy remains independently executable based solely on local observations, without requiring sequential action selection or global coordination at runtime.

In continuous control tasks, the policy is typically modeled as a Dilac distribution (or Gaussian distribution). Without loss of generality, we employ the Dilac policy, which provides deterministic prefix actions $a_{<i}$ given the state during the policy update of agent $i$. This approach not only preserves the flexibility of simultaneous decision-making but also enables efficient parallel pre-training of score models for each agent directly from the dataset. By decoupling the sequential modeling of joint behavior policies from the execution phase, OMSD achieves a unique balance between coordinated learning and decentralized execution, making it highly efficient and scalable for real-world multi-agent scenarios.

While Gaussian policies are standard in continuous control, they are suboptimal for sequential score regularization since sampling stochastic prefix actions causes noise propagation and instability. Instead, we adopt Dilac policies—deterministic mappings with likelihood approximation capacity—to ensure that prefix actions remain stable and deterministic during training.

This design choice aligns with the score distillation requirement and allows high-throughput parallel updates across agents, improving both training efficiency and scalability.

Crucially, OMSD does not employ the diffusion model as an actor network during execution, which could lead to out-of-distribution (OOD) action problems due to the iterative sampling process Mao et al. (2024). Instead, we only perturb the sampled actions from policy $a_i^0 = \pi(a_i|s)$ with a random noise $\epsilon_t \sim \mathcal{N}(\boldsymbol{0}, \boldsymbol{I})$ to construct latent variables $a_i^t$ and use the diffusion model to compute the corresponding score function $\hat{\epsilon}(a_i^t|s, a_{<i}, t)$ as behavior regularization. This approach avoids the computationally expensive ancestral sampling required in denoising steps in traditional diffusion models, significantly accelerating both training and execution.

Figure 3 illustrates the training workflow of OMSD. Joint offline data is reused to train a global Q function $Q^{tot}(s, \boldsymbol{a})$ and agent-wise conditional diffusion models. During policy updates, each agent receives:

- Top-down guidance from $Q^{tot}(s, \boldsymbol{a})$, for identifying high-value regions;
- Bottom-up score regularization from the diffusion model, which conditions on prior agents' actions and regularizes against OOD updates.

This two-way information flow enables coordinated learning while ensuring in-distribution updates at each step. Even when earlier agents' policies deviate, the proper conditional score guides corrections, preserving a stable joint behavior pattern.

Moreover, because diffusion models are used only for score estimation, not sampling, OMSD avoids diffusion-based actor workflows that suffer from iterative sampling inefficiency and OOD action

generation Mao et al. (2024). The final policies remain lightweight, independently executable, and deployable in fully decentralized environments.

## I COMPUTATIONAL RESOURSES

For `MaMuJoCo` and `MPE` experiments, we utilized a single NVIDIA Geforce RTX 3090 graphics processing unit (GPU). For the most complex MaMuJoCo task, training IQL takes 6-10 hours, training the diffusion model for each agent takes 4-6 hours, and training the OMSD policy update only takes 1-2 hours to converge. For the simpler tasks such as MPE and bandit, each module only takes 1 hour and 10 minutes respectively. Since the sequential diffusion model for each agent can be trained in parallel using the data from the dataset, multiple pretraining models can be initiated in parallel to avoid the training time increasing linearly with the number of agents.

## IMPACT STATEMENT

This work advances offline multi-agent reinforcement learning (MARL) by addressing the challenge of unbiased decomposition of multimodal joint action behavior distributions. Our methods improve coordination and decision-making in multi-agent systems, with potential applications in robotics, autonomous vehicles, and collaborative AI systems. By enabling more effective offline learning, our approach reduces the need for risky online exploration in safety-critical domains.

## J USE OF LLMS

We use LLMs for polish writing. Specifically, LLMs assist in refining the grammar, clarity, and overall presentation of the paper, ensuring that the text is clear and professionally written. No experimental results or core content were generated by LLMs.

## K LIMITATION AND FUTURE WORKS

Our current work is limited to continuous control tasks, and we have not yet validated the effectiveness of OMSD on discrete action spaces. In the future, we plan to extend our approach to a wider range of discrete or hybrid tasks to further test its generalizability and practical value.

