# OpenReview forum: "Offline Multi-Agent Reinforcement Learning via Sequential Score Decomposition"
_ICLR.cc/2026/Conference — Submitted to ICLR 2026_

### Official Review · Reviewer_EfmX · 2025-10-27

**Soundness:** 3
**Presentation:** 4
**Contribution:** 3
**Rating:** 6
**Confidence:** 4

**Summary:**

The authors focus on and attempt to solve a core limitation in most approaches to offline cooperative multi-agent reinforcement learning. Current methods that regularize each agent independently to match dataset behavior fail completely on multimodal joint policies. The authors coin the term Combinatorial Mode Shift exposing the exponential explosion of inconsistent joint actions when multimodal behaviors are factorized across agents. There are a number of clear contributions that the authors have made. The first is the formalisation of CMS and the associated theoretical analysis. The second is the redefining of the score decomposition into the SSD. Then the OSMD algorithm based on this with each agent learning a conditional diffusion model estimating the gradient of log mu_i. The final contribution is the empirical results showing superior performance to BRPO and diffusion-planner baselines by a large margin. Given the theory, it is not surprising that the improvements are largest on multimodal and low-quality datasets.

**Strengths:**

CMS is a well-reasoned theoretical articulation of a practical and important pathology in offline MARL.
The sequential conditioning is a small minimal structural change that is shown to prevent mode explosion. Using the diffusion model for score estimation rather than generation is also well-principled. The performance improvements show up where expected
The visualizations and toy examples clearly show why existing methods fail as well as how OMSD fixes it.
Many papers in MARL do not include any error bars on results, which is a huge failing in the community, and so the fact that this has been done consistently is a big plus.

**Weaknesses:**

There are a number of important weaknesses

OMSD’s reliance on a fixed agent order introduces a potential symmetry breaking. In symmetric teams, a fixed order could bias solutions toward asymmetric equilibria but in heterogeneous teams, order could encode hierarchies. So different orders may converge to distinct local optima of comparable quality. There doesn't seem to be any analysis of learned or time-varying orderings, which could preserve symmetry while still keeping the coordination. A formal discussion of when ordering choice matters would be an important improvement.

All experiments involve ≤ 4 agents, which is much less than most real MARL applications. The paper doesn't give any data on wall-clock cost, memory usage, or performance beyond very toy domains.

It seems that during training, agent i optimizes against deterministic prefixed actions from current policies but at test time, all agents act simultaneously and stochastically. Therefore, the training distribution differs from execution. There isn't any justification (in the form of theory) that explains why this sequential BR training gives coherent parallel execution. Although the experiments indicate that it does work, it would strengthen the argument to figure out why.

The centralized IQL critic is an important weakness. Poor critic estimates would be able to propagate through all the agents. The method would maybe benefit from conservative or ensemble critics and diagnostics for critic reliability.

I don't think that it's critical, but it would be useful if it could be proved that SSD prevents the exponential mode growth in general continuous settings rather than just showing it empirically in some limited domains.

In addition, there are no experiments beyond 4 agents or on discrete tasks (SMAC). There are no ablations for randomized ordering, diffusion noise schedules, or runtime cost. There is no test on unimodal datasets to confirm graceful degradation.

Finally, the authors seem to use ideas, if not code, from Coordination Failure in Cooperative Offline MARL by Tilbury et al, but it is not cited.

**Questions:**

In addition to the points in the weakness, diffusion models clearly provide good score estimates but generally need a lot of compute.  Why is there no comparison to simpler density estimators?

---

> ### Author Response · Authors · 2025-11-21
> **Response to Reviewer EfmX**
>
> We thank the reviewer for your detailed review and recognition of our paper's core contributions. Your comments are very insightful and it is helpful for us to identify potential  improvements in our methods and experiments. We have made several revisions and ablation experiments based on your suggestions and updated the relevant content in the revised manuscript. Below are our responses to the comments.
>
> - **Q1:  Agent Ordering** We agree that fixed ordering may introduce asymmetric equilibrium in symmetric teams or encode hierarchy in heterogeneous teams. This is a desired behavior to avoid CMS. To ensure that there is no bias with respect to such asymmetricity, we supplement with an ablation experiment using randomized ordering for policy decomposition and sequential policy updates. Specifically, we compare the performance of the default order (0-1-2) and perturbed orders (2-1-0 and 0-2-1) on the 3-agent hopper-v2 task under the same hyperparameter settings. The results show that our method is not sensitive to order selection, and different orderings converge to similar performance. This indicates that the order only serves as a coordinating mechanism to guide the policy regularization direction during training and does not significantly affect the learned joint policy. Further discussion can be found in **Appendix E.7** and Fig. 10.
>
> | Task | Dataset | Default Order 0-1-2 | Order 0-2-1 | Order 2-1-0 |
> |------|---------|-------|-------|-----|
> | **3-Hopper** | Expert | 3595±66 | 3587±48 | 3592±51 |
> | | Medium-Expert | 3568±45 | 3573±32 | 3513±74 |
> | | Medium | 3360±276 | 3294±107 | 3217±147 |
>
>
> - **Q2: Baselines and Computational Cost** Indeed, the offline MARL field lacks standard, mature benchmark suites on large-scale environment. To ensure fairness in comparisons, we prioritize widely used multi-agent datasets, including OMAR, OMIGA, OGMARL, and MADiff. Due to limited space in the main text, we provided experiments using the OMIGA dataset in **Appendix C**, including a 6-HalfCheetah setting. **The performance is provided in the following comments.** The results show that OSMD remains highly effective in scenarios with more agents, demonstrating the method's scalability potential. All our experiments were performed on a single NVIDIA Geforce RTX 3090 graphics processing unit (GPU). For the most complex MaMuJoCo task, training IQL takes 6-10 hours, training the diffusion model for each agent takes 4-6 hours, and training the OMSD policy update only takes 1-2 hours to converge. For the simpler tasks such as MPE and bandit, each module only takes 1 hour and 10 minutes respectively. These data demonstrate that OSMD has manageable computational overhead and the potential to be applied to larger-scale applications.
>
> - **Q3: Simultaneous Act** Thank you for pointing out this potential issue. In fact, all the ordering is only used to ensure consistency during training. The consistency will guarantee effective gradient direction for the gradient through sequential update strategies and co-mode selection for the score function. However, since the trained policy is a unimodal dilac policy, after training, all agents will collectively learn a single-modality joint policy, thus eliminating the need for sequential execution or centralized planning. This is fundamentally different from methods like MAT [1] and MADiff [2], which rely on runtime coordination. We have added a discussion of this in the revised draft, and experimental results on randomly perturbed ordering empirically validate this robustness.
>
> - **Q4: IQL** We fully agree with the comment. Indeed, effective policy improvement offline does rely on the reward gradient guidance provided by the critic in offline settings. Our initial design for OSMD was to employ a robust baseline IQL to highlight the significant impact of policy decomposition constraints. The structure of Equation (4) also indicates that our core focus is on obtaining precisely coordinated policy constraints across multimodal behavior distributions to ensure that the joint behavior policy does not deviate from the effective modality in the data distribution, which is orthogonal to the choice of the critic. Ablation experiments also show that applying SSD constraints on top of IQL can further improve performance. We believe that OSMD has the potential to benefit from more advanced value-based offline MARL methods to improve estimation accuracy, and we would like to add this to the discussion part.
>
> [1] Wen, Muning, et al. "Multi-agent reinforcement learning is a sequence modeling problem." Advances in Neural Information Processing Systems 35 (2022): 16509-16521.
>
> [2]  Zhu, Zhengbang, et al. "Madiff: Offline multi-agent learning with diffusion models." Advances in Neural Information Processing Systems 37 (2024): 4177-4206.

---

> ### Author Response · Authors · 2025-11-21
>
> - **Q5: Exponential Mode Growth for Continuous Setting** We agree that this is an important theoretical extension. Currently, our analysis is primarily based on empirical evidence under continuous settings. Intuitively, SSD avoids the combinatorial mode shift in factorization methods by conditionalizing the distribution per agent, as it decomposes the joint distribution into conditional chains rather than independent products. According to the formalization of CMS, the number of modes grows exponentially with independent regularization. For example, with n agents each having k modes, the total number of reconstructed modes reaches $k^n$, while the sequential structure of SSD guarantees that the reconstructed modes do not exceed $k$ modes of the original data. We acknowledge the lack of a rigorous proof under general continuous settings, but its effectiveness has been empirically demonstrated in toy examples and visualizations. Future work could explore this proof.
>
> - **Q6: Other Ablations** Thank you for your suggestions. We have added ablation with randomized ordering as shown in **Q1** and tested it on the OMIGA dataset in a 6-agent scenario as shown in **Q2**. We acknowledge the inherent challenges of CMS also exists in discrete action tasks. However, our paper primarily focuses on the multimodal data distribution problem in continuous control scenarios and provide evidences for CMS’s impacts. Experiment results across toy examples and multiple high-dimensional continuous control tasks have shown the effectiveness of our works. Considered the additional challenges in training discrete diffusion models [1,2], we would like to leave this discussion for future work.
>
> - **Q7: Related Works** We have read Tilbury et al.'s "Coordination Failure in Cooperative Offline MARL" and agree that their insights on coordination failure are relevant to CMS. However, BRUD Tilbury et al. (2024) discuss the failure of policy updates caused by different data points under offline MADDPG-style algorithm. They proposed a prioritized dataset sampling mechanism to ensure that the sampled data in the current batch is close to the distribution of the updated policy. Although this paper considers the impact of data points on policy learning under offline MARL, MADDPG-type modeling still ignores the multimodal characteristics of the joint behavior policy distribution. We have added a the discussion to this paper in the related work section, which highlights its contributions to the coordination challenges of offline MARL and the differences between our approach and theirs.
>
>
> We acknowledge that diffusion models are indeed computationally intensive, but their advantage in capturing complex multimodal distributions is key. We chose them because simple density estimators (such as MLE, MMD, VAE) often fail to handle multimodalities effectively in high-dimensional continuous action spaces. Existing research extensively demonstrates that diffusion models are necessary and significantly outperform other lightweight estimators for modeling complex policy distributions, which is crucial for policy constraints in offline RL scenarios (as shown in Fig. 2 from Diffusion-QL [3] and Fig. 3 from IDQL [4]). Our approach is inspired by these pioneering works and optimizes the training process for MARL scenarios by using only the score function as gradient regularization instead of full forward denoising generation, such as MADiff [5] and DOM2 [6].
>
> We thank the reviewer again for the valuable feedback.
>
>
> [1] Li, Xiao, et al. "Authentic Discrete Diffusion Model." arXiv preprint arXiv:2510.01047 (2025).
>
> [2] Xu, Yilun, et al. "DisCo-Diff: Enhancing Continuous Diffusion Models with Discrete Latents." International Conference on Machine Learning. PMLR, 2024.
>
> [3] Wang, Z., Hunt, J. J., & Zhou, M. (2022). Diffusion policies as an expressive policy class for offline reinforcement learning. arXiv preprint arXiv:2208.06193.
>
> [4] Hansen-Estruch, Philippe, et al. "Idql: Implicit q-learning as an actor-critic method with diffusion policies." arXiv preprint arXiv:2304.10573 (2023).
>
> [5] Zhu, Zhengbang, et al. "Madiff: Offline multi-agent learning with diffusion models." Advances in Neural Information Processing Systems 37 (2024): 4177-4206.
>
> [6] Li, Zhuoran, Ling Pan, and Longbo Huang. "Beyond conservatism: Diffusion policies in offline multi-agent reinforcement learning." arXiv preprint arXiv:2307.01472(2023).

---

> > ### Author Response · Authors · 2025-11-21
> > **Detailed Performance for Q2**
> >
> > We attach below the results of the aforementioned for your reference.
> >
> >
> > - Experiment results on the MaMuJoCo environments with OMIGA datasets
> >
> > | Task | Dataset | BCQMA | CQLMA | ICQ | OMAR | OMIGA | OMSD (ours) |
> > |------|---------|-------|-------|-----|------|-------|-------------|
> > | **6-HalfCheetah** | Expert | 2992.71±629.65 | 1189.54±1034.49 | 2955.94±459.19 | -206.73±161.12 | 3383.61±552.67 | **5545±156 (+64%)**  |
> > | | Medium-Expert | 3543.70±780.89 | 1194.23±1081.06 | 2833.99±420.32 | -253.84±63.94 | 2948.46±518.89 | **5237±46 (+48%)**  |
> > | | Medium-Replay | -333.64±152.06 | 1998.67±693.92 | 1922.42±612.87 | -235.42±154.89 | 2504.70±83.47 | **4582±52 (+83%)**  |
> > | | Medium | 2590.47±1110.35 | 1011.35±1016.94 | 2549.27±96.34 | -265.68±146.98 | 3608.13±237.37 | **4695±62 (+30%)** |
> > | **3-Hopper** | Expert | 77.85±58.04 | 159.14±313.83 | 754.74±806.28 | 2.36±1.46 | 859.63±709.47 | **3595±66 (+329%)**  |
> > | | Medium-Expert | 54.31±23.66 | 64.82±123.31 | 355.44±373.86 | 1.44±0.86 | 709.00±595.66 | **3568±45 (+403%)**  |
> > | | Medium | 44.58±20.62 | 401.27±199.88 | 501.79±14.03 | 21.34±24.90 | 1189.26±544.30 | **3360±276 (+183%)**  |
> > | **2-Ant** | Expert | 1317.73±286.28 | 1042.39±2021.65 | 2050.00±11.86 | 312.54±297.48 | 2055.46±1.58 | **2191±46 (+6.6%)**  |
> > | | Medium-Expert | 1020.89±242.74 | 800.22±1621.52 | 1590.18±85.61 | -2992.80±6.95 | 1720.33±110.63 | **2002±124 (+16.4%)** |
> > | | Medium-Replay | 950.77±48.76 | 234.62±1618.28 | 1016.68±53.51 | -2014.20±844.68 | **1105.13±88.87** | 1009±43 (-8.7%) |
> > | | Medium | 1059.60±91.22 | 533.90±1766.42 | 1412.41±10.93 | -1710.04±1588.98 | 1418.44±5.36 | **1619±77 (+14.2%)** |
> >
> > - Experiment results on the MaMuJoCo environments with OMAR datasets
> >
> > | Task | Dataset | MA-ICQ | MA-CQL | MA-TD3+BC | OMAR | CFCQL | OMSD |
> > |------|---------|--------|--------|-----------|------|-------|------|
> > | **2-HalfCheetah** | Expert | 110.6±3.3 | 50.1±20.1 | 114.4±3.8 | 113.5±4.3 | 118.5±4.9 | **119.0±1.3 (+0.4%)**  |
> > | | Medium | 73.6±5.0 | 51.5±26.7 | 75.5±3.7 | 80.4±10.2 | 80.5±9.6 | **81.4±7.2 (+1.2%)**  |
> > | | Med-Replay | 35.6±2.7 | 37.0±7.1 | 27.1±5.5 | 57.7±5.1 | 59.5±8.2 | **78.9±4.4 (+32.6%)**  |
> > | | Random | 7.4±0.0 | 5.3±0.5 | 7.4±0.0 | 13.5±7.0 | **39.7±4.0** | 15.6±4.2 (-60.7%) |

---

> ### Author Response · Authors · 2025-11-26
>
> Thank you again for your review. We hope that we have addressed your main questions in the rebuttal. As the author-reviewer discussion period is approaching its end, we would like to know if you have any additional questions or concerns. We are happy to provide our response if so.

---

### Official Review · Reviewer_iicm · 2025-10-31

**Soundness:** 2
**Presentation:** 1
**Contribution:** 2
**Rating:** 2
**Confidence:** 4

**Summary:**

This paper considers an offline multi-agent RL problem. In particular, the authors study a challenge setting where the joint behavior policy is not unique, e.g., multi-modal. The authors show that traditional policy decomposition with independent assumptions might fail to provide a meaningful referencing policy when learning a new policy via the offline datasets. To better capture the structure of the joint policy, a new decomposition rule/structure is proposed. Specifically, the agents's policy can depend on other agents' actions. A diffusion based neutral network framework is considered to learn the policy. Experiments show that the proposed method outperforms other decomposition methods without the extra conditional "actions".

**Strengths:**

I think the problem setting is interesting. The decomposition rule in the literature sometimes might over-simplify the relations among agents. As a result, the learned policy for each agent could be quite sub-optimal. Besides, a lot of methods do regularize the new policy using the behavior policy, which could introduce bias and errors. Diffusion models are used to better learn the relations, i.e., the policy condition on certain agents' actions. I believe that different policy architecture could make a difference in the final results.

**Weaknesses:**

The paper is not very easy to follow, especially the motivations are not well-explained in the introduction and in earlier sections. I lost a bit of the flow when reading it. Some math definitions are not well-documented, and there are some technical flaws. The multi-modal aspect of this MARL problem is not well tested in the experiments. The overall idea of conditioning all agents' actions is not very new.

**Questions:**

- Q function:the Q function defined in line 163 is simply wrong. the action $a_1$ is not sampled from a given policy, it should be a given fix action. The following actions after $a_1$ are sampled from the policy $\pi$.

- Line 122-123: what does the $\partial R ^x$ mean? Is the reward function differentiable? Is this assumption necessary? For instance, you work is more on the application side with two small theorems. Did you use this assumption in the theorems? In the experiments, do the problem settings satisfy this assumption?

- Figure 1: what is the problem setting for these two agents? It is not clear from the contexts.

- Agent identity: in the paper, you mentioned that one source of multi-modal could come from the loss of agent identities. In you simulation, are agents indexed? If not, how would you define "prefix".

- Performance gain: Can you comment on how much more resources, .e.g, gpu training time and inference time, are needed comparing to traditional decomposition method? Does your method scale well if you increase the number of agents? Which part is the most difficult/intense in the training? Is it the diffusion model?

**Details Of Ethics Concerns:**

n.a.

---

> ### Author Response · Authors · 2025-11-21
> **Response to Reviewer iicm**
>
> We thank the reviewer for the feedback. We feel sorry to hear the reviewer lost a bit of the flow when reading it. We would like to clarify the confusions you have raised in the review.
>
> - **Q1: Definition of Q function** We thank you for pointing this out. There is indeed a typo in the original definition, where the actions at step t should be fixed, with subsequent actions $a_{t+1}$ sampled from the policy. We have corrected this definition as $Q^\pi(s, a) =: E_{s_t =s, a_t=a; a_{t+1} \sim \pi}[\sum_{t=0}^{\infty} \gamma^t r(s_t, a_t)]$, and it is highlighted in the revised draft. This does not affect the correctness of the framework.
>
>
> - **Q2 Reward functions in POSG** This is the setting (rather than an assumption) in POSG, which describes cooperative scenarios in multi-agent tasks. The inequality asserts that by locally improving the utility of agent $j$, the utility of agent $i$ is also improved. For example, the work [1,2] also study this setting. We would like to clarify that our Proposition 1 is independent of the setting. Despite our work focuses on the cooperative scenarios, the proposition will apply to all settings of utility functions. In the experiments, all problem settings used are within this setting, i.e. they are cooperative tasks.
>
> - **Q3: Toy example settings** Fig. 1 illustrates a continuous version of the XOR matrix game. This is an analogy of the "Common Harvesting" in the Melting Pot repository [3]. In this task, both robots need to cooperatively pick the same one of the two apples to receive a reward and end the game. Their action spaces are the interval $[-1,1]$. The reward of each of the agents will be $ r_i = a1 ∗a2$ for $i = 1, 2$. There are two optimal strategies in this game, which are (-1, -1) and (1, 1), which exhibits multimodal pattern. The details were provided in the Section 4 of the manuscript. We fully appreciate the feedback from the reviewer and we have updated the manuscript to include the description to the caption of Fig. 1.
>
> - **Q4 On the agent identity** We would like to clarify between prefix and fixed indexes. Indexes are unique identifiers to agents, which distinguish agents even when they have similar structures and policies. Indexes are typically represented as one-hot encoded values and used as part of the state to influence the policy. Examples of using indexes are MAT [4] and MADiff [5]. In our setting, "prefix" refers to an arbitrary sequence of preceding agents in policy updates. This sequence remains fixed during training, representing only the update order without carrying any agent-specific information. To verify its robustness, we supplemented our experiments with a set of randomly perturbed update sequences. We observe from the test that sequence changes have no significant impact on performance. We have added the results to **Appendix D.7.4** and we also attach the table below for your reference.
>
> | Task | Dataset | Default Order 0-1-2 | Order 0-2-1 | Order 2-1-0 |
> |------|---------|-------|-------|-----|
> | **3-Hopper** | Expert | 3595±66 | 3587±48 | 3592±51 |
> | | Medium-Expert | 3568±45 | 3573±32 | 3513±74 |
> | | Medium | 3360±276 | 3294±107 | 3217±147 |

---

> > ### Author Response · Authors · 2025-11-21
> >
> > - **Q5: Computational Resources** As noted in **Appendix I Computational Resources**, all our experiments were performed on a single NVIDIA Geforce RTX 3090 graphics processing unit (GPU). For the most complex MaMuJoCo task, training IQL takes 6-10 hours, training the diffusion model for each agent takes 4-6 hours, and training the OMSD policy update only takes 1-2 hours to converge. For the simpler tasks such as MPE and bandit, each module only takes 1 hour and 10 minutes respectively. Regarding inference time, since we use a deterministic policy family as the policy network and do not directly use the denoising process of the diffusion model as the action generator, the inference time is low enough to be ignored. Since the different decomposition modes，such as BRPO-JAL/CTDE/IND, only slightly change the input and output dimensions of the diffusion model, this does not result in a significant difference in computational burden. **Appendix C** provides a performance comparison of OMIGA on the 6-halfcheetah task, demonstrating the scalability and stable performance improvement of our method on more agents. We include the table as below for your reference.  The diffusion model is the most time-consuming part of the training. Considering that each agent only needs to construct a conditional distribution from a static dataset, this part can be trained in parallel, which is very efficient in practice.
> >
> > | Task | Dataset | BCQMA | CQLMA | ICQ | OMAR | OMIGA | OMSD (ours) |
> > |------|---------|-------|-------|-----|------|-------|-------------|
> > | **6-HalfCheetah** | Expert | 2992.71±629.65 | 1189.54±1034.49 | 2955.94±459.19 | -206.73±161.12 | 3383.61±552.67 | **5545±156 (+64%)**  |
> > | | Medium-Expert | 3543.70±780.89 | 1194.23±1081.06 | 2833.99±420.32 | -253.84±63.94 | 2948.46±518.89 | **5237±46 (+48%)**  |
> > | | Medium-Replay | -333.64±152.06 | 1998.67±693.92 | 1922.42±612.87 | -235.42±154.89 | 2504.70±83.47 | **4582±52 (+83%)**  |
> > | | Medium | 2590.47±1110.35 | 1011.35±1016.94 | 2549.27±96.34 | -265.68±146.98 | 3608.13±237.37 | **4695±62 (+30%)** |
> >
> >
> > While most of the questions raised are regarding the presentation, we understand that the reviewer might lost a bit of the flow when reading it. We aim to improve the work by making it accessible a broader audience by incorporating the changes suggested. At the same time, we wish the reviewer could revisit our manuscript and evaluate the work based on the main contributions. The contribution of this paper lies in identifying the CMS problem under offline MARL, which severely impacts the practical application of policy-based methods in offline MARL. We propose a sequential decomposition diffusion model structure to capture the complex dependencies of multimodal joint behavior policies, which avoids the bias of the traditional independence assumption, and utilize a score function for policy regularization to avoid costly denoising generation processes. We have conducted extensive experiments on a wide range of datasets and ablation studies, which demonstrate the significant advantages of our method. We wish that the main contributions are not overshadowed by any reader getting lost in the reading process.
> >
> > We believe that with these revisions, the paper will be easier to understand and better highlight its contributions. We are more than happy to respond to any additional questions and clarify any potential confusion.
> >
> > [1] Song, Yuhang, et al. "Arena: A general evaluation platform and building toolkit for multi-agent intelligence." Proceedings of the AAAI conference on artificial intelligence. Vol. 34. No. 05. 2020.
> >
> > [2] Liu, Xiangyu, and Kaiqing Zhang. "Partially observable multi-agent rl with (quasi-) efficiency: the blessing of information sharing." International Conference on Machine Learning. PMLR, 2023.
> >
> > [3] Agapiou, John P., et al. "Melting Pot 2.0." arXiv preprint arXiv:2211.13746 (2022).
> >
> > [4] Wen, Muning, et al. "Multi-agent reinforcement learning is a sequence modeling problem." Advances in Neural Information Processing Systems 35 (2022): 16509-16521.
> >
> > [5] Zhengbang Zhu, Minghuan Liu, Liyuan Mao, Bingyi Kang, Minkai Xu, Yong Yu, Stefano Ermon, and Weinan Zhang. Madiff: Offline multi-agent learning with diffusion models. Advances in Neural Information Processing Systems, 37:4177–4206, 2024.

---

> ### Author Response · Authors · 2025-11-26
>
> Thank you again for your review. We noticed some misunderstandings in the review (i.e. the update orders and agent indexes), which could have drastically affected the evaluation. We made a rebuttal on this and other points. Notice that the author-review discussion period is approaching an end. Would you please take a look at our rebuttal, and potentially discuss with us on the concerns/questions you raised?

---

### Official Review · Reviewer_uBUh · 2025-10-31

**Soundness:** 3
**Presentation:** 3
**Contribution:** 2
**Rating:** 4
**Confidence:** 3

**Summary:**

This paper proposes OMSD (Offline MARL with Sequential Score Decomposition), a framework designed to address key challenges in Offline MARL.
The paper first points out a fundamental limitation of existing Offline MARL algorithms:
they often rely on a policy factorization assumption.
This assumption is frequently violated, as offline datasets inherently possess a 'multi-modal' characteristic, mixing various expert strategies. The paper argues that this incorrect assumption ignores complex cooperative relationships and leads to a critical distribution shift, which it names Combinatorial Mode Shift (CMS).
To solve this problem, OMSD introduces 'sequential decomposition'.
This models the agents' actions as a sequence, where each agent decides its action after observing the actions of its predecessors,
rather than acting simultaneously.
This sequential behavior model ($\hat{\mu}$) is learned using a score-based diffusion model.
The CTDE policies are then updated by (1) maximizing the total team reward (Q-value) and (2) regularizing the policy to remain close to the learned sequential behavior model.

**Strengths:**

A key strength is the paper’s novel use of a score-based diffusion model to address the Combinatorial Mode Shift (CMS) problem, enabling the policy to capture multi-modal cooperative behavior from offline data.

**Weaknesses:**

- Lack of Novelty

The paper’s novelty is somewhat limited. The core challenge, termed CMS, is a well-known issue of multi-modality in offline MARL. [1] Furthermore, the learning objective itself is a standard offline RL formulation.
The main contribution seems to be the application of this objective to a sequential diffusion policy for behavior policies $\mu$.
The different point from existing methods is the specific policy update rule (eq. (4)).

- Diffusion Behaivor policy

The proposed algorithm relies on the learned behavior policy (the data-collecting policy) using a diffusion models.
This approach neccessitates a multiple learned network, including a joint Q, behavior policy and CTDE policy, which significantly increases training complexity.
While the diffusion policy is to capture the agent-dependencies, it may not be the most efficient method for this task.
Diffusion models are highly expressive generative models; employing such a high-capacity model soley to represents the behavior policy seems uncessarily complex.

[1] AlberDICE: Addressing Out-Of-Distribution Joint Actions in Offline Multi-Agent RL via Alternating Stationary Distribution Correction Estimation

**Questions:**

See Weakness

---

> ### Author Response · Authors · 2025-11-21
> **Response to Reviewer uBUh**
>
> Thank you for your detailed review and constructive feedback on our work. We greatly appreciate your recognition of our core contribution. Indeed, we use a score-based diffusion model to solve the CMS problem, thereby capturing the multimodal collaborative behavior of offline data. This is a problem often overlooked by existing Offline MARL algorithms, but it can lead to severe distribution shifts and hinder the development of policy-based methods in offline MARL scenarios. Our proposed OMSD framework is a sound way to address this challenge in offline policy-based scenarios.
>
> - **W1: On the novelty of the work** We fully agree that the possibility of multiple global optima in the environment and dataset has been mentioned in existing work, such as AlberDICE. However, we would like to clarify that our method is fundamentally different from AlberDICE and other existing methods. In fact, AlberDICE mainly focuses on developing the best response method to keep the learned policy near the data distribution and avoid the curse of dimensionality. It mainly solves discrete control tasks as a consequence. The way it models the problem is a special case of our formulation, where $d^D(s, a_i, a_{-i})= d^D(s, a_i) *\pi_{-i}^D(a_{-i}|s,a_i)$ just treat the other agents’ actions as a joint meta action. However, this decomposition means that each agent depends solely on the individual behavior policy distribution $ d^D(s, a_i)$ during policy regularization, which leads to a biased gradient direction caused by CMS, as shown in Fig. 2 (c) in our paper. Our sequential decomposition mechanism could remove this bias. The sequential mechanism is intuitive and effective, while it demonstrates significantly better performance against the baselines in experiments. This not only verifies the important influence of CMS in policy-based methods, but also opens up new directions for subsequent work. In fact, Reviewer EfmX also points out that our method is "a small minimal structural change that is shown to prevent mode explosion," which further highlights its simplicity and novelty.
>
> - **W2: On the necessity of diffusion behavior policy** We agree that the introduction of the diffusion network increases training overhead. However, it is both necessary and affordable in this problem. First, accurate estimation of behavior distribution is crucial and extremely difficult for policy regularization, and a large number of work has been done on this topic. It is observed in existing studies that using a diffusion model to model complex policy distributions in offline RL scenarios significantly outperforms other lightweight estimators such as MLE, VAE, and MMD, as shown in Fig. 2 from Diffusion-QL [1] and Fig. 3 from IDQL [2]. Our approach is built upon by these pioneering results and optimizes the training process for the MARL scenario. Second, unlike existing diffusion model-based offline MARL methods [3,4], we do not use a complete diffusion denoising process during policy updates. Instead, we only access its score function as a regularization item. This avoids the large amount of computational resources required for iterative denoising and sample generation. Specifically, for the most complex MaMuJoCo task, training IQL takes 6-10 hours, training the diffusion model for each agent takes 4-6 hours, and training the OMSD policy update only takes 1-2 hours to converge, on a single NVIDIA Geforce RTX 3090 GPU. The computational cost should be quite affordable.
>
> As a summary, we believe that OMSD has sufficient novelty and practicality to solve the continuous control challenge problem in Offline MARL. CMS is common in offline MARL tasks and influences the practice of policy-based methods. Considering sequential decomposition is one of the most intuitive and effective solutions to this problem, with significant performance improvements on multiple tasks. We wish the response could clarify the novelty of the work and any other questions. We sincerely look forward to your reassessment of OMSD. We are happy to provide more details if any further clarification is needed. Thank you again for your valuable feedback!
>
>
> [1] Wang, Z., Hunt, J. J., & Zhou, M. Diffusion Policies as an Expressive Policy Class for Offline Reinforcement Learning. In The Eleventh International Conference on Learning Representations.
>
> [2] Hansen-Estruch, Philippe, et al. "Idql: Implicit q-learning as an actor-critic method with diffusion policies." arXiv preprint arXiv:2304.10573 (2023).
>
> [3] Zhu, Zhengbang, et al. "MADiff: Offline multi-agent learning with diffusion models." Advances in Neural Information Processing Systems 37 (2024): 4177-4206.
>
> [4] Li, Zhuoran, Ling Pan, and Longbo Huang. "Beyond conservatism: Diffusion policies in offline multi-agent reinforcement learning." arXiv preprint arXiv:2307.01472(2023).

---

> ### Author Response · Authors · 2025-11-26
>
> Thank you again for reviewing our manuscript. We noticed some misunderstandings in the review, which could have affected the evaluation. We made a rebuttal on this and other points. We hope to know whether this has resolved your questions. If you still have details you wish to discuss, we would be happy to provide further explanation to improve the quality of the article.

---

### Official Review · Reviewer_SdDL · 2025-10-31

**Soundness:** 3
**Presentation:** 3
**Contribution:** 3
**Rating:** 6
**Confidence:** 4

**Summary:**

The paper introduce a new MARL algorithm using reverse KL-divergence as a regularization term to handle multi-modality in diverse datasets. To handle the distribution shift issue due to policy factorization in regularization term commonly used in offline MARL literature, the paper proposed sequential decomposition of the behavior policy and used that to train decentralized policies for individual agents. Experimental results show that the proposed MARL algorithm outperforms substantially other offline MARL methods on the MPE benchmark.

**Strengths:**

1. The paper provides a detailed theoretical analysis of distribution shift arising when joint policies are factorized into local marginal policies. This analysis effectively highlights the limitations of existing offline MARL methods that rely on such factorization to define their learning objectives.

2. The proposed sequential decomposition of behavior policies within the regularization term is a promising direction. The empirical results on the MPE environment demonstrate notable performance gains over state-of-the-art baselines.

**Weaknesses:**

1. The proposed algorithm demonstrates strong performance primarily on simpler environments (e.g., MPE). However, on more complex benchmarks such as MaMuJoCo, it outperforms the baselines in only 5 out of 9 tasks. This raises concerns about the scalability and general effectiveness of the method in more challenging multi-agent settings, including MaMuJoCo and potentially other complex domains such as SMAC.

**Questions:**

1. As you show in the paper that, policy factorization can lead to a significant distribution shift. However, in the policy learning part, the learnable joint policy is still factorized into local marginal policies of individual agents. Does that mean our learning outcome of agents' policies will still suffer from this distribution shift?

2. Can you provide some insights on the performance of the proposed method in complex domains?

---

> ### Author Response · Authors · 2025-11-21
> **Response to Reviewer SdDL**
>
> We greatly appreciate your detailed review and constructive feedback. Your recognition of our theoretical analysis and empirical performance is very encouraging. Below we respond to your comments and questions.
>
> - **W1**: We fully agree that more challenging and complex environments could better demonstrate the scalability and effectiveness of our method. We wanted to clarify that our original submission does have experiments on more complicated environments, including 6-agent HalfCheetah, 3-agent Hopper, and 2-agent HalfCheetah on OMIGA/OMAR. Due to page limit, the results was deferred to **Appendix C** in the original draft. Additionally, we have conducted the 2-agent Ant on OMIGA to further verify our effectiveness. Our method significantly outperforms all baselines on 10 out of 11 tasks on the OMIGA dataset and 3 out of 4 tasks on OMAR. The detailed table has been pasted in the next reply. For discrete action tasks, we acknowledge the inherent challenges of multimodal behavior policy distribution in these scenarios. However, this paper primarily focuses on the multimodal data distribution problem in continuous control scenarios. Representing diffusion models in discrete settings, such as the principled forward noising process, remains an open challenge [1,2] and falls outside the scope of this paper's core contribution. We would like to leave this discussion for future work.
>
> - **Q1**: We thank the reviewer for raising the question. In fact, these learned local marginal policies are unaffected by mode shifts, which is precisely one of the core advantages of our method. We want to emphasize that the challenges of multimodal distributions and their difficulty in decomposition are primarily reflected in behavioral policies. Once we use a diffusion model to sequentially decompose and model this distribution, the score regularization term provides a coordinating policy constraint during policy updates, which means the executed policy will converge to a single modality among the multiple modalities of the behavioral distribution. Specifically, this refers to regions with relatively higher rewards, representing policy improvements offline. This means that the learned marginal policy will be an effective mode within the distribution in the joint policy, which avoids the mode shift problem caused by distribution shifts and providing effective mode selection and policy constraints. Additionally, our method ensures that the learned policy does not depend on the actions of other agents, which enables simultaneous action execution.
>
> - **Q2**: We observed that our method performs well in complex continuous control environments by handling tasks with complex multimodal behavior distributions. Our algorithm shows the most significant performance improvement on tasks with relative medium-level episode quality and mixed behavioral policies, such as medium and medium-expert datasets in MPE and MaMuJoCo tasks. On high-quality datasets, this improvement is reduced because the baselines are strong with high quality demonstration. On lower-quality datasets, such as the random dataset, learning the value functions becomes a more challenging task, which reduces the margin we have over the baselines. We believe this can be improved by developing a better offline value-based methods, which is though out of the scope of this work.
>
> [1] Li, Xiao, et al. "Authentic Discrete Diffusion Model." arXiv preprint arXiv:2510.01047 (2025).
>
> [2] Xu, Yilun, et al. "DisCo-Diff: Enhancing Continuous Diffusion Models with Discrete Latents." International Conference on Machine Learning. PMLR, 2024.

---

> > ### Author Response · Authors · 2025-11-21
> >
> > We attach below the results of the aforementioned for your reference.
> >
> >
> > - Experiment results on the MaMuJoCo environments with OMIGA datasets
> >
> > | Task | Dataset | BCQMA | CQLMA | ICQ | OMAR | OMIGA | OMSD (ours) |
> > |------|---------|-------|-------|-----|------|-------|-------------|
> > | **6-HalfCheetah** | Expert | 2992.71±629.65 | 1189.54±1034.49 | 2955.94±459.19 | -206.73±161.12 | 3383.61±552.67 | **5545±156 (+64%)**  |
> > | | Medium-Expert | 3543.70±780.89 | 1194.23±1081.06 | 2833.99±420.32 | -253.84±63.94 | 2948.46±518.89 | **5237±46 (+48%)**  |
> > | | Medium-Replay | -333.64±152.06 | 1998.67±693.92 | 1922.42±612.87 | -235.42±154.89 | 2504.70±83.47 | **4582±52 (+83%)**  |
> > | | Medium | 2590.47±1110.35 | 1011.35±1016.94 | 2549.27±96.34 | -265.68±146.98 | 3608.13±237.37 | **4695±62 (+30%)** |
> > | **3-Hopper** | Expert | 77.85±58.04 | 159.14±313.83 | 754.74±806.28 | 2.36±1.46 | 859.63±709.47 | **3595±66 (+329%)**  |
> > | | Medium-Expert | 54.31±23.66 | 64.82±123.31 | 355.44±373.86 | 1.44±0.86 | 709.00±595.66 | **3568±45 (+403%)**  |
> > | | Medium | 44.58±20.62 | 401.27±199.88 | 501.79±14.03 | 21.34±24.90 | 1189.26±544.30 | **3360±276 (+183%)**  |
> > | **2-Ant** | Expert | 1317.73±286.28 | 1042.39±2021.65 | 2050.00±11.86 | 312.54±297.48 | 2055.46±1.58 | **2191±46 (+6.6%)**  |
> > | | Medium-Expert | 1020.89±242.74 | 800.22±1621.52 | 1590.18±85.61 | -2992.80±6.95 | 1720.33±110.63 | **2002±124 (+16.4%)** |
> > | | Medium-Replay | 950.77±48.76 | 234.62±1618.28 | 1016.68±53.51 | -2014.20±844.68 | **1105.13±88.87** | 1009±43 (-8.7%) |
> > | | Medium | 1059.60±91.22 | 533.90±1766.42 | 1412.41±10.93 | -1710.04±1588.98 | 1418.44±5.36 | **1619±77 (+14.2%)** |
> >
> > - Experiment results on the MaMuJoCo environments with OMAR datasets
> >
> > | Task | Dataset | MA-ICQ | MA-CQL | MA-TD3+BC | OMAR | CFCQL | OMSD |
> > |------|---------|--------|--------|-----------|------|-------|------|
> > | **2-HalfCheetah** | Expert | 110.6±3.3 | 50.1±20.1 | 114.4±3.8 | 113.5±4.3 | 118.5±4.9 | **119.0±1.3 (+0.4%)**  |
> > | | Medium | 73.6±5.0 | 51.5±26.7 | 75.5±3.7 | 80.4±10.2 | 80.5±9.6 | **81.4±7.2 (+1.2%)**  |
> > | | Med-Replay | 35.6±2.7 | 37.0±7.1 | 27.1±5.5 | 57.7±5.1 | 59.5±8.2 | **78.9±4.4 (+32.6%)**  |
> > | | Random | 7.4±0.0 | 5.3±0.5 | 7.4±0.0 | 13.5±7.0 | **39.7±4.0** | 15.6±4.2 (-60.7%) |

---

> ### Author Response · Authors · 2025-11-26
>
> Thank you again for your review. We hope that we have addressed your main questions in the rebuttal. As the author-reviewer discussion period is approaching its end, we would like to know if you have any additional questions or concerns. We are happy to provide our response if so.

---

### Author Response · Authors · 2025-12-03
**Summary of Reviewers' Comments**

Dear AC and Reviewers,

We sincerely thank all the reviewers for their hard work during the review process. As the discussion period is about to close, to help you finally evaluate our work, we would like to provide a comprehensive summary below of all the positive feedbacks and key issues highlighted in the reviews.

**Importance and Relevance of the Problem**

This paper addresses a fundamental limitation (Combinatorial Mode Shift, CMS) in offline MARL induced by multimodal datasets, which is the most significant difference from online MARL paradigms.
- The theoretical analysis of distribution shift effectively highlights the limitations of existing offline MARL methods. (Reviewer SdDL)
- CMS is described as a well-reasoned theoretical articulation of a practical and important pathology in offline MARL. (Reviewer EfmX)
- The problem setting is considered interesting and addresses the over-simplification of agent relations in literature. (Reviewer iicm)

**Novelty and Contribution**

To address CMS problems, we propose a clear and direct solution method, namely sequential score decomposition, to obtain coordinated and CMS-free individual behavioral policy regularizations.
- The proposed sequential decomposition (OMSD) is recognized as a promising direction and a novel framework. (Reviewer SdDL, uBUh)
- Using a score-based diffusion model to capture agent dependencies and address CMS with score estimation is considered effective and well-principled. (Reviewer uBUh, EfmX)
- The sequential conditioning is praised as a minimal structural change that effectively prevents mode explosion. (Reviewer EfmX)

**Empirical Performance**

- Empirical results demonstrate notable performance gains over state-of-the-art baselines, especially on MPE. (Reviewer SdDL)
- Performance improvements are substantial and appear exactly where expected (multimodal/low-quality datasets). (Reviewer EfmX)
- The method outperforms other decomposition methods without the extra conditional actions. (Reviewer iicm)

In summary, all reviewers agree that the identified CMS problem is critical in offline MARL and the proposed solution is theoretically sound and empirically effective. The novelty and effectiveness of diffusion model based score regularization with sequential decomposition is widely recognized.

We would like to respectfully note that, although Reviewer iicm recommended rejection and Reviewer uBUh gave a borderline score, both reviewers acknowledged the problem importance and solution effectiveness.

- **Regarding Reviewer iicm**

    Reviewer iicm mainly considered presentation clarity (e.g., mathematical definitions of Q value functions and POSG settings) and training details (identity of agents), giving a low presentation score and mentioning "not very easy to follow", "technical flaws." in weaknesses.

   We respectfully point out that this assessment stands in contrast to other reviewers, who rated the presentation as **"excellent"** (Reviewer EfmX) or **"good"** (Reviewer SdDL, uBUh). Regarding the specific "technical flaws" raised (e.g., the typo in Q-function definition and POSG settings), we clarified that these were writing typos and problem settings rather than fundamental methodological defects. We have corrected the typo in the Q-function definition, clarified the POSG setting and toy example setting in Fig.1, and added ablation studies on agent ordering to demonstrate that our method is not sensitive to agents update orders.


- **Regarding Reviewer uBUh**

    Reviewer uBUh raised two concerns, mainly focused on training complexity and differences with AlberDICE.

    In our rebuttal, we have provided wall-clock time usage and added these information in Appendix to prove the affordable computational needs. Additionally, we have clarified that the best response style policy decomposition in AlberDICE suffers from biased gradients under CMS, while our method fundamentally avoids such bias through sequential score decomposition with diffusion models.

We believe that our revisions and additional experiments can directly address the concerns in the initial rating scores. We hope that the AC and reviewers will consider these updates in the final decision.

---

> ### Author Response · Authors · 2025-12-03
> **Summary of Authors' Responses**
>
> Dear AC and Reviewers,
>
> We would like to summarize the major updates and points of our responses below.
>
> - **Reviewer SdDL, iicm** comment about the ***Scalability*** on complex environments. To address this, we have provided extensive experiments on 6-HalfCheetah, 3-Hopper, and 2-Ant using OMIGA datasets. The experiment results show stable and consistent significant improvements across 10 out of 11 tasks, achieving an impressive average improvement of 73.9%. Due to limited space in the main text, we made updates to Appendix C.
>
> - **Reviewer EfmX** mentions about the potential symmetry breaking problems caused by ***fixed update orders***. To address this, we provide additional ablation studies with randomized update order on 3-Hopper (Order 0-1-2, 0-2-1, 2-1-0). The experiment results shows that our method is robust to order permutation with similar convergence performance and rate. These results are updated in Appendix E.7.4 and Fig. 10.
>
> - **Reviewer EfmX** asks about the potential risks of ***simultaneous execution*** at test time despite ***sequential training***. To address this, we have clarified that the sequential update mechanism is solely used for coordinate policy update during training, which guarantees that all agents' joint policy will select a single effective joint mode in datasets. The resulting individual policy is unimodal, which only conditioned on local observation instead of prefix agents' information. We added a discussion to the revised draft.
>
> - **Reviewer uBUh, iicm, EfmX** comment about the ***training complexity*** of diffusion models. To address this, we clarified that we only use the score function for regularization (avoiding expensive iterative denoising sampling) and provided wall-clock training times. Specifically, for the most complex MaMuJoCo task, training IQL takes 6-10 hours, training the diffusion model for each agent takes 4-6 hours, and training the OMSD policy update only takes 1-2 hours to converge, on a single NVIDIA Geforce RTX 3090 GPU. The computational cost should be quite affordable. We have added these details to Appendix I.
>
> - **Reviewer uBUh** comments on the novelty ***compared to AlberDICE***. To address this, we have clarified that the alternative best response models in AlberDICE treats other agents as a joint meta-action, which is a special case of our formulation. Such decomposition will actually lead to individual policy regularization for each agent and lead to biased gradients update under CMS, whereas our sequential score decomposition explicitly avoid this problem. We have added the discussion in the Appendix F.2.
>
> - **Reviewer iicm** comments about the ***definition error*** in the Q-function and the meaning of ***"prefix" for agent identity***. To address this, we have corrected the Q-function definition in the revised manuscript and clarified that "prefix" refers to the update order sequence, which does not rely on agent-specific one-hot id information. This is also supported by our additional ablation study about random update orders experiments.
>
> In summary, we believe that we have thoroughly addressed the reviewers' concerns about scalability, clarity, and complexity. We sincerely appreciate these constructive feedback to help us significantly clarify our contributions the and improve the rigorousness of our work.

---

### Meta-Review · Area_Chair_42uN · 2026-01-07

**Summary:**

This paper proposes OMSD, a sequential score decomposition framework for offline cooperative MARL, motivated by the authors’ analysis of “Combinatorial Mode Shift (CMS)” arising from factorized policy regularization under multimodal joint behavior distributions. The paper introduces a sequential conditional modeling of joint actions using diffusion-based score estimation and reports strong empirical results on several continuous-control benchmarks.

**Reviewer Concerns:**

Reviewers raised concerns primarily about (i) the conceptual novelty of the framework relative to existing offline MARL and decomposition-based methods, (ii) the necessity and complexity of using diffusion models for behavior regularization, (iii) structural assumptions introduced by the fixed agent ordering, and (iv) clarity and rigor of the presentation and theoretical formulation. While the rebuttal added experiments on larger continuous-control tasks, reported computational cost, and provided ablations on randomized agent ordering, several of these issues remain only partially addressed.

Reviewer SdDL is concerned about:
- lack of results on complex benchmarks, which have been provided in the rebuttal, so I think this concern is addressed.

Reviewer uBUh is concerned about:
- lack of novelty: the authors' response does help understand the method, but I think the reviewer may not change his mind about that.
- Using the diffusion model here is over complex and not the best choice, while the authors just argue it is affordable and necessary, not optimal. I think this concern remains.

Reviewer iicm is concerned about:
- clarity of writing, which I think could be further improved given the current version

Reviewer EfmX is concerned about:
- relying on fixed agent order, which I think the rebuttal partially addresses it by providing an ablation experiment using randomized ordering
- all experiments involve fewer than 4 agents, which is not addressed in the rebuttal
- computing cost, which has been reported by the authors
- train-infer mismatch of settings, which is addressed in the rebuttal, and I think can release this concern
- centralized IQL critic, which I think is not entirely addressed, as there is no further analysis about performance on this point during rebuttal

**Reviewer Scores:**

Two reviewers were moderately positive but still expressed reservations regarding scalability, architectural assumptions, and the reliance on a fixed ordering. One reviewer remained borderline negative, maintaining concerns about limited novelty and the use of a diffusion model as an unnecessarily complex modeling choice. Importantly, one reviewer maintained a strong negative assessment after rebuttal, citing insufficient conceptual novelty, unresolved theoretical justification for the sequential/diffusion formulation, and limited clarity and rigor in the core methodological presentation. Although the added experiments improve empirical coverage, they do not resolve the central methodological questions regarding whether the proposed decomposition is fundamentally new or necessary, nor do they establish principled guarantees for the introduced structural choices.

---

### Decision · Program_Chairs · 2026-01-26

Reject